# A coronene-based semiconducting two-dimensional metal-organic framework with ferromagnetic behavior

Renhao Dong[1], Zhitao Zhang[2], Diana C. Tranca[1], Shengqiang Zhou[2], Mingchao Wang[1], Peter Adler[3], Zhongquan Liao[4], Feng Liu [5], Yan Sun[3], Wujun Shi[3], Zhe Zhang[1], Ehrenfried Zschech[4], Stefan C.B. Mannsfeld[1], Claudia Felser[3] & Xinliang Feng[1]

Metal–organic frameworks (MOFs) have so far been highlighted for their potential roles in catalysis, gas storage and separation. However, the realization of high electrical conductivity ($>10^{-3}$ S cm$^{-1}$) and magnetic ordering in MOFs will afford them new functions for spintronics, which remains relatively unexplored. Here, we demonstrate the synthesis of a two-dimensional MOF by solvothermal methods using perthiolated coronene as a ligand and planar iron-bis(dithiolene) as linkages enabling a full $\pi$-$d$ conjugation. This 2D MOF exhibits a high electrical conductivity of ~10 S cm$^{-1}$ at 300 K, which decreases upon cooling, suggesting a typical semiconductor nature. Magnetization and $^{57}$Fe Mössbauer experiments reveal the evolution of ferromagnetism within nanoscale magnetic clusters below 20 K, thus evidencing exchange interactions between the intermediate spin S = 3/2 iron(III) centers via the delocalized $\pi$ electrons. Our results illustrate that conjugated 2D MOFs have potential as ferromagnetic semiconductors for application in spintronics.

[1] Department of Chemistry and Food Chemistry & Center for Advancing Electronics Dresden, Technische Universität Dresden, 01062 Dresden, Germany. [2] Helmholtz-Zentrum Dresden-Rossendorf, Institute of Ion Beam Physics and Materials Research, Bautzner Landstr. 400, 01328 Dresden, Germany. [3] Max Planck Institute for Chemical Physics of Solids, 01187 Dresden, Germany. [4] Fraunhofer Institute for Ceramic Technologies and Systems (IKTS), 01109 Dresden, Germany. [5] State Key Laboratory for Mechanical Behavior of Materials, Xi'an Jiaotong University, Xi'an 710049, China. Correspondence and requests for materials should be addressed to X.F. (email: xinliang.feng@tu-dresden.de)

The realization of high electrical conductivity and long-range magnetic ordering within a single material is highly appealing to meet the demands for data storage and processing[1,2]. Typically, ferromagnetic semiconductors are of considerable interest for spintronics (spin electronics) applications[3–5], where the electronic spins can be modulated by external electrical and magnetic fields to perform logic operations and act as memory. Currently, materials employed as ferromagnetic semiconductors generally comprise inorganic solid compounds like Heusler compounds and dilute magnetic semiconductors[4,6–8] and organic/molecular film semiconductors[9–16]. In these inorganic solids, direct metal–metal bonds or indirect interactions among metal atoms mediated by small ligands (such as oxo atoms) enable the long-range electronic communication, which is essential to achieve the electrical conductivity and magnetic ordering[4,6–8]. Nevertheless, their entirely inorganic composition prevents a facile tunability of magnetic and conductive properties due to the lack of both ligand functionalization and structural diversity. In the case of molecular film semiconductors, their electronic and magnetic properties can be tuned to a much higher degree, due to the structural diversities of organic monomers and functional groups that can be employed. Notably, the molecular semiconductors such as vanadium-tetracyanoethylene[11] and metal-phthalocyanine[12–14] have exhibited strong magnetic coupling at high temperature, which are considered as an emerging class of spin transport media with long spin lifetime due to their carbon-based light-atom compositions. Nevertheless, the low mobility and complex transport properties in molecular semiconductors still hinder their practical applications for spintronics.

In contrast, metal–organic frameworks (MOFs) are hybrid porous materials based on crystalline coordination polymers consisting of metal atoms or clusters connected by organic ligands[17,18]. Their properties and functions can be tuned by varying abundant organic ligands, metal centers and framework geometries, thus enabling their potential in catalysis[19], sensing[20], gas storage[21,22] and separation[23], etc. However, it is still challenging to realize magnetic MOF semiconductors. The conventionally reported three-dimensional (3D) MOFs showed extremely low bulk electrical conductivity ($10^{-12}$ to $10^{-8}$ S cm$^{-1}$)[20,24] and weak magnetic coupling due to the large separation of metal centers by multi-atoms, insulating organic ligands, etc. Recent advances disclosed that the incorporation of conducting guests in MOFs could significantly enhance the electrical conductivity[25,26]. Moreover, immobilizing redox-active ligands with mixed-valences, such as 2,5-dihydroxybenzoquinone, into the backbones of iron(III) complexes could generate long-range charge delocalization and strong magnetic exchange[27–29], leading to high conductivity ($\sim10^{-4}$ to $\sim0.2$ S cm$^{-1}$) and high-temperature magnetic ordering (Curie temperature even could reach 105 K). In addition, the design of conjugated 2D MOFs by employing planar organic ligands and square-planar metal-complex linkages inducing full delocalization of $\pi$-electrons has also led to improved electrical conductivity ($>10^{-3}$ S cm$^{-1}$)[20,24]. These reported 2D MOFs are based on benzene[30–32] or triphenylene-derivatives[33–36] with thiol[30,31,35,36], hydroxyl[33] or amino[32,34] groups linked via Ni or Cu metal centers and have been integrated as electrode materials for applications in transistor[37], electrocatalysis[35,38,39], chemiresistive sensor[40] and energy storage[41]. Despite the magnetic behavior of these 2D MOFs remains unexplored, high delocalization of $\pi$-electrons in a 2D plane enables the interaction between charge carriers and localized spins to drive magnetic ordering[3,42], which makes conjugated 2D MOFs as intriguing candidates for ferromagnetic semiconductors[42–44].

In this study, we demonstrate a 2D MOF (PTC-Fe) consisting of polycyclic aromatic hydrocarbon monomer, namely 1,2,3,4,5,6,7,8,9,10,11,12-perthiolated coronene (PTC), as ligand connected by planar iron-bis(dithiolene) linkage enabling full $\pi$-$d$ conjugation in a 2D plane. Thus, such 2D MOF is featured with hexagonal lattices and van der Waals layer-stacking structure. Four-probe van der Pauw electrical measurement revealed the room temperature conductivity value of $\sim10$ S cm$^{-1}$ for bulk compressed pellet. A variable-temperature conductivity measurement displayed a non-linear increase of conductivity with temperature, indicating a typical semiconducting behavior. A density functional theory (DFT) calculation was carried out to estimate the band gap as $\sim0.2$ eV for a monolayer MOF. A variable-temperature magnetic susceptibility measurement as well as $^{57}$Fe Mössbauer spectra demonstrated that the PTC-Fe exhibited ferromagnetic ordering within nanoscale magnetic clusters at low temperatures (below $\sim20$ K). Our work highlights conjugated 2D MOFs as a class of conductive materials exhibiting ferromagnetic and semiconducting features for potential spintronics application.

## Results

**Synthesis and structure.** Black polycrystalline PTC-Fe 2D MOFs (Fig. 1a) were synthesized from reaction of 1,2,3,4,5,6,7,8,9,10,11,12-perthiolated coronene (PTC)[45] with ammoniacal solutions of iron acetate (Fe(OAc)$_2$) in the mixture of deoxygenated water and dimethylformamide (DMF) heated at 120 °C in a sealed vial for 48 h. The solid product was collected and washed by deoxygenated DMF, dilute hydrochloric acid (0.1 M), water and acetone and dried under vacuum at 100 °C. The attenuated total reflection IR (ATR-IR) spectra of PTC-Fe (Supplementary Fig. 1) displayed that the S-H signals at 2512 cm$^{-1}$ disappeared in the PTC-Fe, suggesting that the thiol groups in monomer PTC were efficiently coordinated to Fe atoms. Thermogravimetric analysis (TGA) revealed that PTC-Fe decomposed above 300 °C, as evidenced by the pronounced weight losses above this temperature (Supplementary Fig. 2). The nitrogen sorption isotherms were measured at 77 K to evaluate the porosity of PTC-Fe (Supplementary Fig. 3), which showed gas uptake and release during the adsorption and desorption processes. The Brunauer–Emmett–Teller specific surface area was calculated to be 210 ($\pm5$) m$^2$/g with the existence of counter ions of NH$_4^+$ in the pores.

Powder X-ray diffraction (PXRD) measurements with Co Kα irradiation ($\lambda = 1.79$ Å) reveal a highly crystalline structure in PTC-Fe with prominent (100), (200) and (210) peaks at $2\theta = 8.8°$, 17.6°, and 21.7°, respectively, as shown in Fig. 1b. We simulated several possible stacking arrangements for the layers of PTC-Fe by DFT calculation (Supplementary Fig. 4). The experimentally resolved pattern agrees well with the AB packing model with 25% shifting in X and Y directions between neighboring layers. Thus, the PXRD results combined with the theoretical simulation provide a structural model of PTC-Fe (Fig. 1b and enlarged image shown in Supplementary Fig. 4d), which exhibits a hexagonal pattern within the $ab$ planes and a AB layer stacking along the $c$ direction. The square planar, conjugated iron-bis(dithiolene) linkages extend the $\pi$-electron conjugation of coronene in a 2D plane. From the peak at $2\theta = 8.8°$, we can infer a hexagonal unit cell with $a = b = \sim11.7$ Å. The peak at $2\theta = 26.0°$, corresponding to the (001) reflection suggests an ordered stacking with an interlayer distance of $\sim3.9$ Å.

Transmission electron microscopy (TEM) studies provide further evidence of 2D long-range order and layer stacking in PTC-Fe. The TEM images show a non-porous morphology in PTC-Fe (Fig. 1c and Supplementary Fig. 5). In-plane periodicity is determined by selected area electron diffraction (SAED, inset in Fig. 1c), which presents strong polycrystalline diffraction rings. The ring size reveals an ordered network with $a = \sim1.2$ nm cell size. High resolution TEM (HRTEM) images clearly present

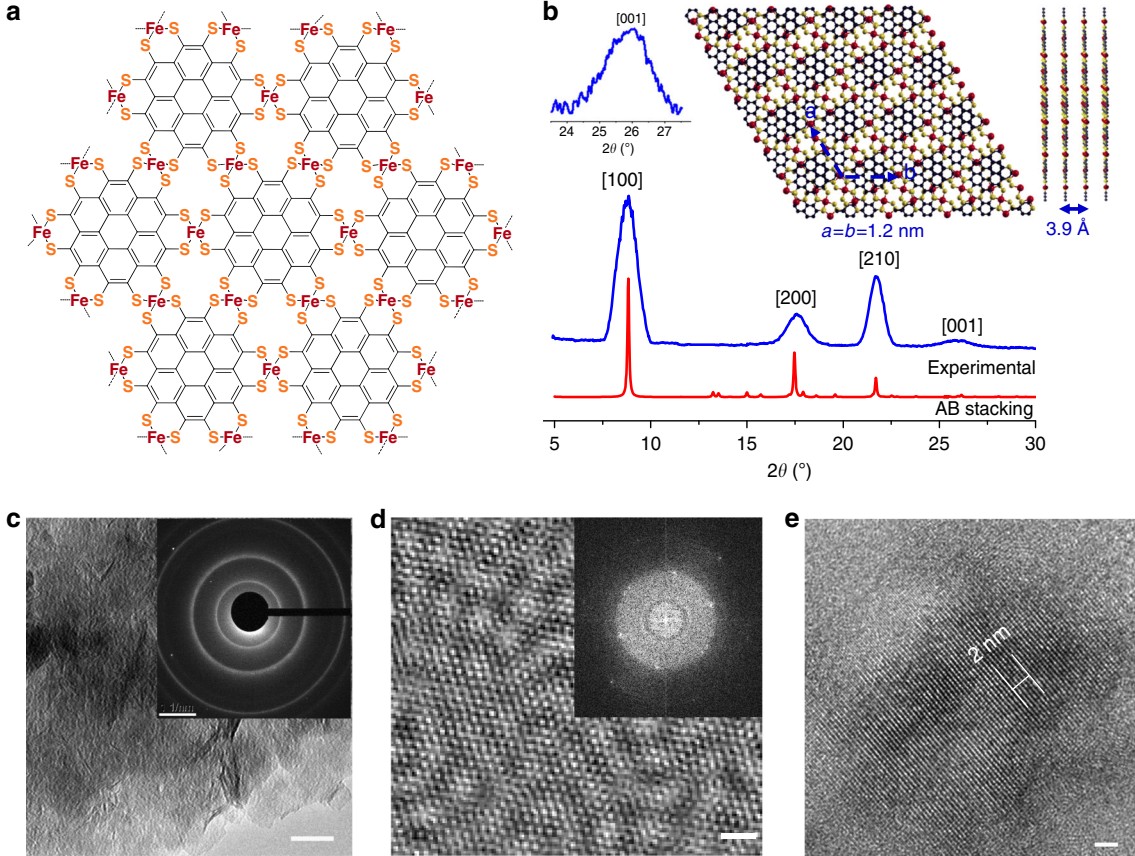

**Fig. 1** Structural characterizations of PTC-Fe 2D MOF. **a** Schematic structure of PTC-Fe. **b** Experimental and simulated PXRD patterns. Insets: enlarged experimental PXRD peak corresponding to (001) reflection and the crystal structure simulation of AB stacking model with 25% shifting in X and Y directions. **c** transmission electron microscopy (TEM) image. Scale bar=50 nm. Inset: Selected area electron diffraction (SAED) pattern. Scale bar=1 nm$^{-1}$. **d** high-resolution in-plane TEM images. Scale bar=2 nm. Inset: fast Fourier transform (FFT) pattern from image (**d**). **e** high resolution TEM image from the side view. Scale bar=2 nm. The distance across five lattices is 2 nm

crystalline domains within a dozen of nanometers (Fig. 1d and Supplementary Fig. 5). Fast Fourier transform (FFT) analysis of a crystalline domain in Fig. 1d further demonstrates a hexagonal structure with a lattice parameter $a = b = ~1.2$ nm, in excellent agreement with PXRD and DFT results. A cross view of PTC-Fe sample shows the layer-stacking structure with an interlayer distance of ~4 Å (Fig. 1e).

**Compositional characterization.** Next we investigated the composition of PTC-Fe. X-ray photoelectron spectroscopy (XPS) spectrum discloses the presence of Fe 2p, N 1 s, C 1 s, S 2 s, and S 2p core levels (Supplementary Fig. 6a). The N signal originates from the counter ions of $NH_4^+$ balancing the charged system. The high-resolution Fe 2p photoemission spectrum shows two sets of peaks (Supplementary Fig. 6b), with binding energies of 710.9 and 723.9 eV, which are assigned to the 2p3/2 and 2p1/2 levels, respectively, indicating the presence of Fe(III) species. The S 2p peak in the XPS spectrum occurs at a binding energy of ~162 eV. Deconvolution of the S 2p signal generates high-intensity dual peaks at 161.5 and 162.7 eV derived from the Fe–S units, evidencing efficient complexation between Fe ions and thiol groups (Supplementary Fig. 6c). Quantitative analysis of the Fe and S signals provides a Fe:S ratio of ~0.98:4, which is in consistence with the expected PTC-Fe model.

Synchrotron powder X-ray absorption spectroscopy (XAS) was employed to characterize the coordination geometry and iron valence in PTC-Fe (Fig. 2a and Supplementary Methods). As a reference, the known one-dimensional coordination polymer (TTB-Fe)[46] comprising 1,2,4,5-tetrathiolbenzene and iron (III), as well as Fe foil, FeO and $Fe_2O_3$ inorganic solids were also investigated by XAS. Thus, the Fe K-edge X-ray absorption near edge structure (XANES) spectra of PTC-Fe reveal the same coordination geometry as the TTB-Fe. Figure 2b shows the Fourier transform of the $\kappa$-weighted extended X-ray absorption fine structure (EXAFS) spectra of PTC-Fe, as well as $Fe_2O_3$ and TTB-Fe as contrast samples. The EXAFS spectra present a predominant peak in PTC-Fe, which is originated from the nearest-neighboring sulfur coordination shell around the Fe atoms. Based on this peak, Fe-S distance was calculated to be ~2.23 Å. Besides this first-shell interaction, the second-shell atomic interactions were also observed, based on which distance of Fe–C of ~3.39 Å was calculated (Supplementary Fig. 7). These bond lengths are very close to those of the contrast sample TTB-Fe, which are well consistent with the iron-bis(dithiolene) geometry. However, another contrast sample $Fe_2O_3$ exhibits two predominant peaks at ~1.44 Å and ~2.57 Å, which arise from Fe–O and Fe–Fe bonds, respectively. Therefore, the XANES and EXAFS spectra of PTC-Fe and the contrast experiments provide strong proof on the formation of square planar iron-bis(dithiolene) complexes via the coordination of PTC and Fe ions. Moreover, no metal oxides such as FeO and $Fe_2O_3$ were detected in the PTC-Fe.

In order to further investigate the iron coordination geometry, local electronic structure, and magnetic properties, $^{57}Fe$ Mössbauer spectra were measured between 300 and 5 K (Fig. 2c, d, Supplementary Methods, Supplementary Fig. 8 and

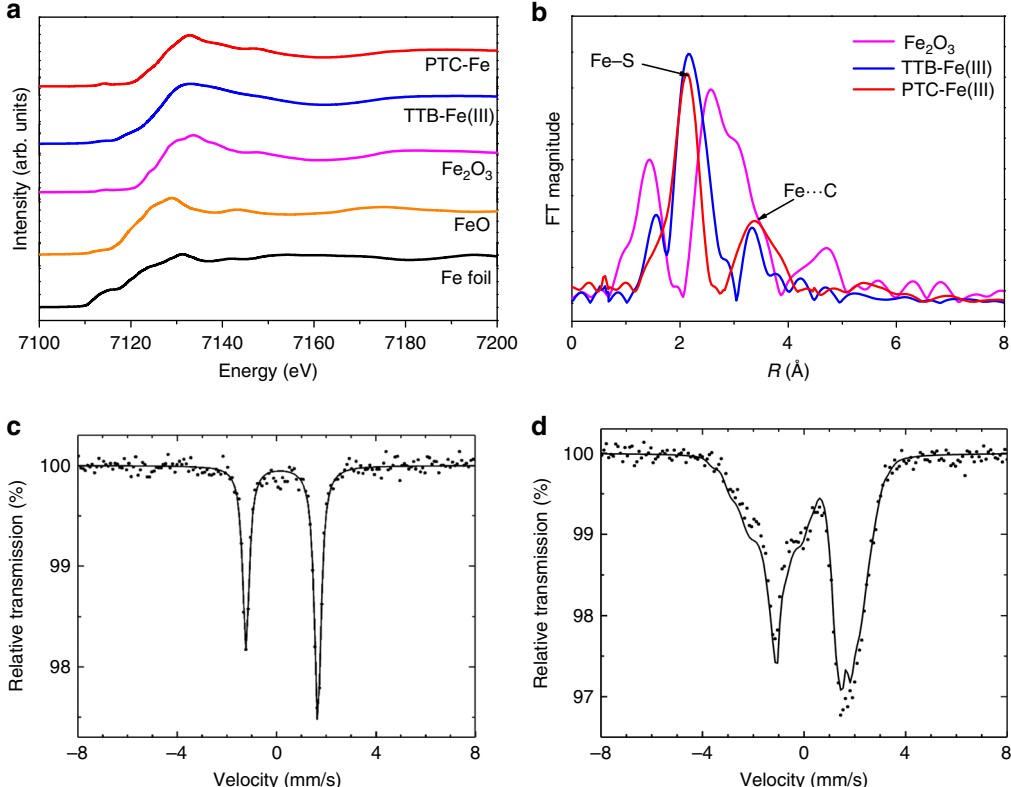

**Fig. 2** Compositional characterizations of PTC-Fe. **a** Normalized XANES spectra at the Fe K-edge of PTC-Fe and its reference compounds, including Fe foil, FeO, $Fe_2O_3$, and 1,2,4,5-tetrathiolbenzene-Fe(III) (TTB-Fe(III)). For the PTC-Fe, the energy of adsorption edge (E0) suggests that the valence state of the Fe ion in the PTC is 3+. **b** Fourier transform of the EXAFS at Fe K-edge of PTC-Fe as well as $Fe_2O_3$ and TTB-Fe as the contrast samples. **c, d** [57]Fe Mössbauer spectra at 294 K and 5 K, respectively. Dots and black line correspond to experimental data and calculated spectrum, respectively

Supplementary Table 1). Spectra obtained by cooling down from 300 to 10 K mainly feature a single quadrupole doublet. The isomer shift IS = 0.21 mm/s and quadrupole splitting QS = 2.89 mm/s at $T$ = 300 K are within the range of values typical for Fe (III) complexes with thiolate ligands and square planar coordination geometry[47,48], which is in agreement with the anticipated Fe-bis(dithiolene) coordination in PTC-Fe. The intensity asymmetry of the two doublet components is a texture effect reflecting preferred orientation of crystallites in the sample. The d-orbital splitting pattern in square planar ligand environment is known to stabilize an intermediate spin S = 3/2 state for $Fe^{3+}$, and the large quadrupole splitting reflects the unequal electron population of the Fe 3d orbitals split by the ligand field (valence contribution to the electric field gradient)[27,49]. The S = 3/2 (intermediate spin) ground state of $Fe^{3+}$ is a consequence of the square planar coordination geometry, which leads to a pronounced energetic stabilization of the $3dz^2$ orbital with respect to the $dx^2-y^2$ orbital, and the energy of the $dz^2$ orbital is comparable to the energies of the $dxy$, $dxz$, and $dyz$ orbitals. Compared with the Fe(III), somewhat larger isomer shifts are expected for square planar Fe(II) complexes with an intermediate spin S = 1. The temperature dependence of IS reflects the usual second-order Doppler shift. A broadened quadrupole doublet (Supplementary Fig. 8) is observed at 10 K; however, at 5 K, the Mössbauer spectrum exhibits magnetic hyperfine splitting suggesting a magnetic ordering transition (Fig. 2d). Furthermore, Mössbauer spectra at low temperature also prove no iron oxides purities in the PTC-Fe.

**Electronic structure of PTC-Fe.** We measured solid-state UV-Vis spectrum of PTC-Fe (Supplementary Fig. 9). Importantly, the

electronic absorption features of PTC-Fe extend well into the near-infrared (NIR) range. Such low-energy electronic excitations are common in highly conjugated organic/metal-organic and conducting polymers[26]. In order to estimate the electronic band structure of PTC-Fe, DFT calculations were performed (Fig. 3, Supplementary Methods and Supplementary Figs. 10–12). The density of states (DOS) describe the type and the number of states occupied at a certain energy and are essential for determining the carrier concentrations and energy distributions within a semiconductor[50]. In this respect, we provide the DOS results for both the single layer (Fig. 3a, b, Supplementary Figs. 10, 11) and multilayer of the 2D MOF with AB stacking model (Fig. 3c, d, Supplementary Fig. 12). Spin up, spin down and mixed spin directions have been considered in the DOS calculation. The band structure simulation for the single layer indicates a band gap of ~0.2 eV (Fig. 3a, b), while that for the AB stacking system exhibits a metallic character (Fig. 3c, d). In addition, the single layer and AB stacking models present a band gap of ~0.7 eV and ~1 eV for the spin-up channel, respectively (Fig. 3b, d). However, for spin down channel, a semi-conducting feature remains for single layer while a metallic character is dominantly presented for AB stacking. Layer stacking is believed to narrow the band gap for most of the 2D materials[51]. In addition, due to the negatively charged structure in PTC-Fe, a compensating cation ($NH_4^+$) has been introduced into the cell to avoid electrostatic potential and energy diverge induced by net charge, which could also play a crucial role in narrowing the band gap. Around the Fermi level the dominating states are coming from Fe-d and S-p orbitals (Supplementary Fig. 10a and c). Analysis of the projected density of states (PDOS) near to the Fermi level show a hybridization of the p-orbitals of S and the $dyz$- and $dxz$-orbitals of Fe, suggesting that the p-orbitals of the system are delocalized (Supplementary

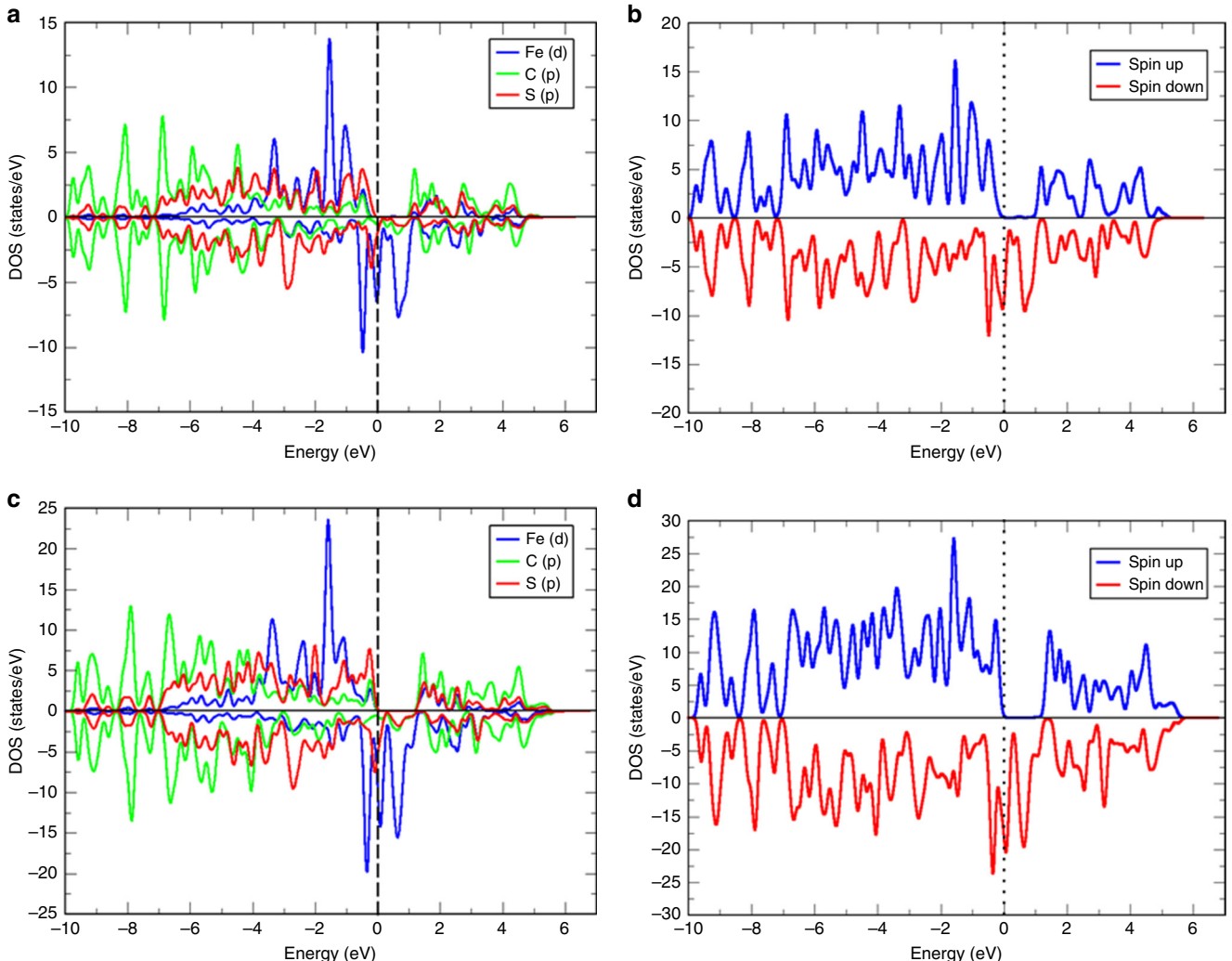

**Fig. 3** Electronic band structure of PTC-Fe near the Fermi level. **a** Total density of states (DOS) for a single layer. **b** Spin up/Spin down for a single layer. **c** Total DOS for AB stacking. **d** Spin up/Spin down for AB stacking

Fig. 10b and d). The bulk materials are expected to be semi-metallic in the *ab* directions (due to the metal ions and the S atoms) and semiconducting in the *c* direction due to weak van der Waals interactions.

**DC conductivity measurements**. The bulk electrical properties of PTC-Fe were determined from pelletized samples with the thickness of ~0.3 mm obtained by pressing powders at ~1 GPa (Inset in Fig. 4a). The *I–V* curves were measured in the van der Pauw geometry and displayed Ohmic response between −1.0 and +1.0 V (Supplementary Fig. 13), which gave an electrical conductivity of the pellets as high as ~10 S cm$^{-1}$ at 300 K. This is among the highest values for the thus far reported intrinsically conducting MOF powders (Supplementary Table 2)[20,24,32].

The variable-temperature conductivity measurements showed a non-linear increase of conductivity from 20 to 320 K (Fig. 4a), indicating a typical semiconducting feature. The natural logarithm of conductivity (ln$\sigma$) plotted versus reciprocal temperature (1/T) is presented in Fig. 4b. The plot shows a linear region over a temperature range of 125–300 K, consistent with a thermally activated transport dominant in this temperature range. The slope of the linear region in Fig. 4b corresponds to the activation energy ($E_a$), which can be extracted from the following Arrhenius law:

$\sigma \sim \exp(E_a/kT)$, where $\sigma$ is the conductivity, $E_a$ is the activation energy, $k$ is the Boltzmann constant ($8.617 \times 10^{-5}$ eV/K) and $T$ is the temperature[31]. The value of $E_a$ was evaluated from the slope of ln$\sigma$ versus 1/T plot and was calculated as 0.2 eV. As shown in the inset of Fig. 4b, the plot of ln($\sigma$) versus $T^{-1/4}$ over the temperature region 65–130 K, which is well fitted to the Mott variable range hoping (Mott–VRH) model[52]. We ascribe this hopping process to the grain boundaries between the crystallites dominating the temperature dependence of conductivity in the bulk polycrystalline pellets, giving rise to apparent semiconducting behavior[32].

**Magnetic properties of PTC-Fe**. Iron-based MOFs [Fe(III)] can exhibit magnetic exchange interactions between neighboring iron atoms through the bridging ligands[27,29]. In order to probe the magnetic properties of the PTC-Fe, we measured the magnetization using a superconducting quantum interference device-vibrating sample magnetometer (SQUID-VSM) (Fig. 5).

Figure 5a and Supplementary Fig. 14a show the variation of the magnetization with applied magnetic field (*H*) measured at different temperatures. While at 50 K the magnetization increases linearly with field, as expected for a paramagnetic system, a sigmoidal shape of *M*(*H*) below ~20 K signals the onset of

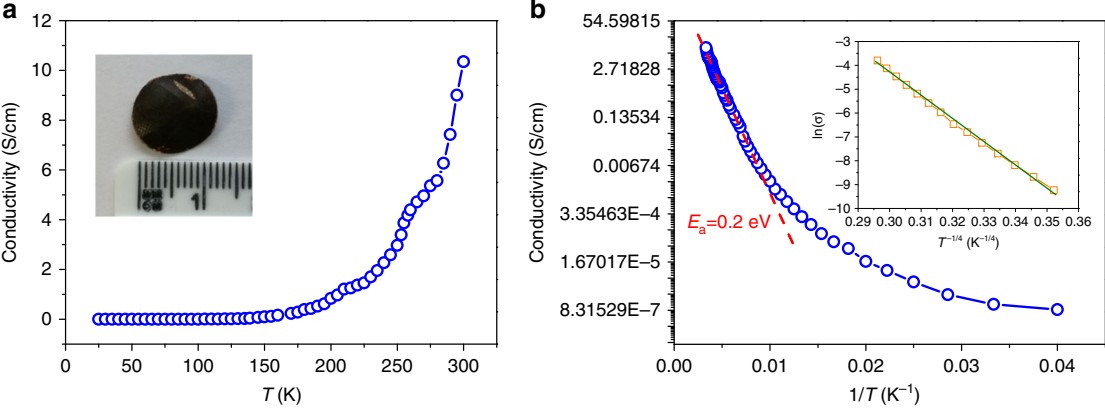

**Fig. 4** Electrical conductivity of PTC-Fe 2D MOFs. **a** Electrical conductivity ($\sigma$) as a functional of temperature ranging from 30 K to 300 K. **b** Plot of ln $\sigma$ versus the reciprocal of the temperature ($1/T$). Inset: Plot of ln ($\sigma$) versus $T^{-1/4}$ over the temperature region 65–130 K

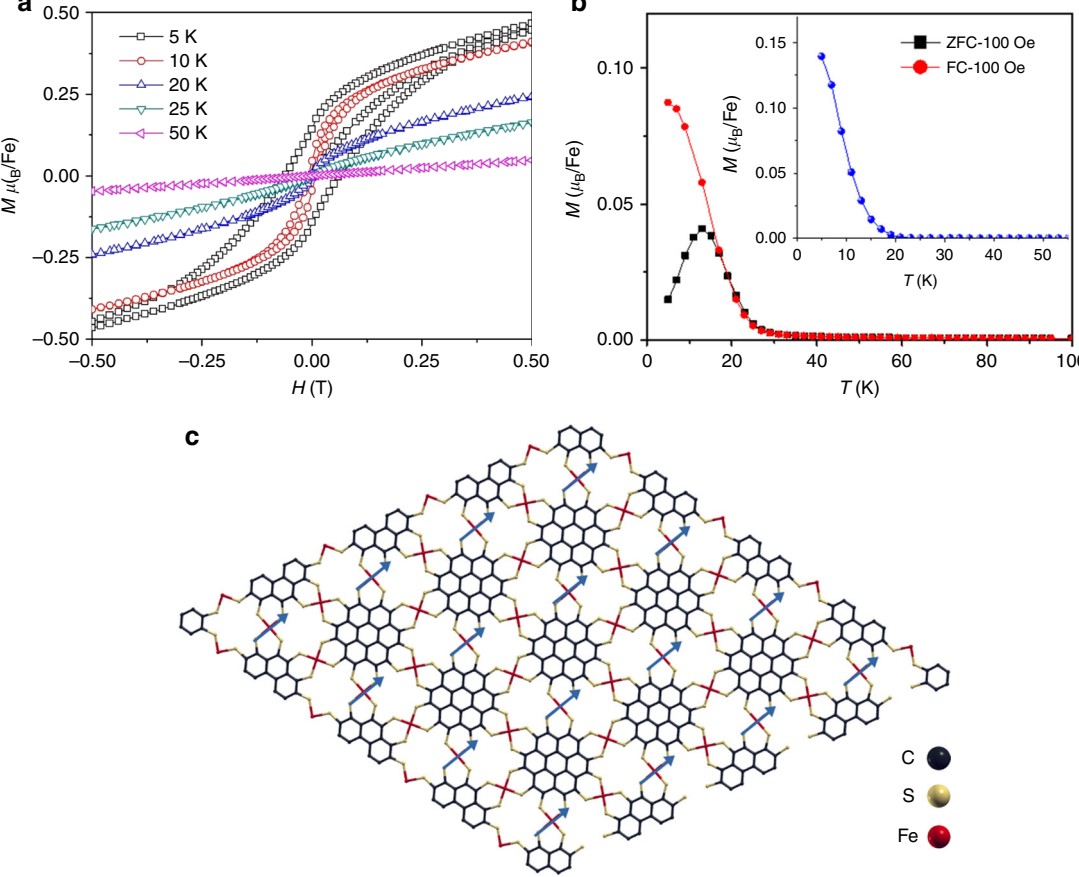

**Fig. 5** Magnetic properties of PTC-Fe. **a** Magnetizations as functions of applied magnetic field (H) measured at different temperatures. **b** Magnetization of PTC-Fe as a function of temperature measured in a 100 Oe field under field-cooled (FC) and zero-field-cooled (ZFC) conditions. Inset: Temperature dependent remanent magnetization. **c** ferromagnetic ground state of PTC-Fe

ferromagnetism. For temperatures $T \leq 10$ K a hysteresis is evident and the hysteresis width increases with decreasing temperature (coercive field ~0.07 T at 5 K). Furthermore, the $T$-dependent magnetization data (Fig. 5b) feature an increase in $M$ below 20 K and a cusp in the zero-field cooled (ZFC) magnetization near 15 K. Below 15 K the FC and ZFC data diverge. These observations point to nanoscale ferromagnetism with an average blocking temperature of about 15 K[53]. The evolution of ferromagnetism below ~20 K is in agreement with the variable-temperature

magnetic remanence measurements (inset in Fig. 5b). These clues are in accordance with the Mössbauer spectra (Supplementary Fig. 8) where line broadening near 10 K signals the onset of spin freezing on the Mössbauer time scale (~$10^{-8}$ s), while the broad hyperfine pattern at 5 K verifies that the majority of the sample is in a frozen magnetic state. However, the very broad hyperfine field distribution (Supplementary Fig. 8, bottom) suggests a wide distribution of relaxation times due to a variation in the size of the magnetic nano-domains.

## Discussion

To understand the origin of the relatively high-temperature (<20 K) ferromagnetic semiconducting ground state in PTC-Fe, we investigated the magnetic properties of the organic ligand PTC as well as coronene as contrast samples by SQUID. The temperature dependent remanent magnetization plots reveal that the ligand PTC and the building core coronene contribute ignorable magnetic ordering even at 5 K while the coordination complexes comprising PTC and Fe behave ferromagnetism (Supplementary Fig. 15). We then evaluated the spin density distribution in AB-stacking PTC-Fe by DFT calculation (Supplementary Fig. 16), which suggests that the Fe atoms provide the major spins compared with the C and S atoms, associated with localized $d$ electrons[44]. In this case, Fig. 5c shows the magnetic ground states of the PTC-Fe with unit-cell magnetization, in which the magnetic moments are predominantly localized on the iron atoms. Notably, the magnetic ground state of the system at 0 K will be ferromagnetic if the magnetic exchange energy is positive whereas anti-ferromagnetic if negative. Here, an exchange energy of $E_{ex} = 1.22$ meV ($E_{ex} = E_{AFM} - E_{FM}$) has been obtained by DFT calculation, further revealing the intrinsic ferromagnetic character of this 2D MOF[43,54]. The simulated Curie temperature of PTC-Fe is ~16 K by employing Ising model (calculation seen in Supplementary Methods)[54], which further supports the SQUID measurements on the magnetic ordering with a blocking temperature of ~15 K (Fig. 5b).

With the preliminary determination of spin distribution, we attempt to further achieve an understanding how the magnetic coupling appears between the localized spins on Fe toms. Given the distance between the neighboring lateral Fe atoms from ~0.54 to ~1.20 nm in PTC-Fe MOF (Fig. 1b and Supplementary Fig. 7), direct exchange between the $d$ orbitals of lateral Fe is relatively weak, and thus cannot be responsible for the strong ferromagnetic coupling between the intralayer Fe atoms. In addition, the layer distance is ~0.38 nm and the distance between the neighboring interlayer Fe atoms is ~0.5 nm, also leading to relatively weak direct exchange interaction. Thus, we propose that the magnetic coupling is induced by indirect exchange interaction[14,27–29,43,44,55,56]. Namely, the localized spin moments of the Fe atoms polarize the delocalized $p$ orbital through exchange interaction. That is because this layer-stacked PTC-Fe MOF comprises of fully conjugated planar structure, enabling strong hybridization between the $d/p$ orbitals of Fe, the coronene core, and the Fe-bis(dithiolene) nodes, which has a critical impact on the magnetic properties of the lattice. DOS plots also indicate that $d_{xz}/d_{yz}$ orbitals of Fe participate in the hybridization of $p$ orbitals (Supplementary Figs. 10 and 12), leading to strong exchange interaction between the localized $d$ orbitals and the $p$ orbitals, and hence ferromagnetic coupling. This indirect exchange through delocalized electrons is consistent with the Zener $d$-$p$ exchange[42,57] or the Ruderman–Kittel–Kasuya–Yosida (RKKY)[58,59] exchange mechanism.

In summary, we have synthesized a coronene-based conjugated 2D MOF that exhibits record electrical conductivity of ~10 S cm$^{-1}$ for bulk compressed pellets at room temperature. The variable-temperature conductivity measurements revealed a typical semiconducting behavior for PTC-Fe. The variable-temperature magnetic susceptibility measurements indicated that PTC-Fe exhibits a ferromagnetic ground state at low temperature resulting from the unique hybridization between the $d/p$ orbitals of Fe, the coronene core, and the Fe-bis(dithiolene) nodes. Our work indicates that endowing 2D MOFs with strongly delocalized $\pi$ systems can be an effective strategy to develop novel ferromagnetic MOF semiconductors. The possible contribution from the magnetic impurities to the ferromagnetic ordering has been excluded via the combined analysis of XPS, XANES, EXAFS, and

$^{57}$Fe Mössbauer spectra. It should be noticed that the conductive and magnetic behavior presented here did not rule out the impact of the defects possibly generated from grain boundaries, the crystallite tilting and the edges in the bulk materials[60], due to the polycrystalline feature and structural complexity as a result of heterogeneity. In addition, the tuning of the defects generated from the uncoordinated vacancy sites was able to lead to long-range ferromagnetic coupling in MOFs, which have been significantly proved via theoretical and experimental approaches[61]. Our results hopefully encourage more physical studies on the ferromagnetic and semiconducting properties as well as spintronic applications, relying on the growth of MOF single-crystals or 2D crystalline films with tuned thickness and enlarged crystalline domain size by interfacial synthesis or chemical vapor deposition methods, which remains to be explored.

## Methods

**Materials.** The ligand, 1,2,3,4,5,6,7,8,9,10,11,12-perthiolated coronene (PTC), was synthesized according to our previous reported protocols[45]. Starting from coronene, 1,2,3,4,5,6,7,8,9,10,11,12-dodecachlorocoronene was prepared by chlorination reaction. Nucleophilic replacement of all peripheral chlorosubstituents was achieved using lithium benzylthiolate at room temperature, which afforded 1,2,3,4,5,6,7,8,9,10,11,12-dodecakis(benzylthio)coronene as a red powder in 62% yield. After reductive cleavage of the protective benzyl groups under Birch conditions using lithium in anhydrous liquid ammonia at −78 °C, the dodecalithiocoronene-1,2,3,4,5,6,7,8,9,10,11,12-dodecathiolate was obtained. The subsequent direct treatment with aqueous hydrogen chloride and hydrogen peroxide afforded persulfurated coronene (PSC) in 61% isolated yield. The reduction of PSC by NaBH$_4$ could afford PTC via the cleavage of S–S bonds.

**Synthesis of PTC-Fe 2D MOFs.** A degassed solution of 0.014 mmol (10.0 mg) of PTC in 0.5 mL of DMF, 0.1 mL of aqueous ammonia (NH$_4$OH, 6 M), and a degassed solution of anhydrous Fe(OAc)$_2$ (7.5 mg, 0.043 mmol) in 0.5 mL of water were transferred into a 10 ml glass vial and sonicated for 5 min at 0 °C. Then, the vial was sealed and heated at 120 °C in an oven for 48 h, followed by natural cooling to room temperature. The solid product was afterwards collected and washed three times by deoxygenated DMF, dilute HCl (0.1 M), water and acetone. After dried at 100 °C under vacuum for 24 h, ~12 mg of black crystals was obtained and stored in Ar. Anal. Calcd. for [Fe$_3$C$_{24}$S$_{12}$]$^{3-}$ •(NH$_4^+$)$_3$: C, 32.19; S, 42.92; N, 4.69; H, 1.34. Found: C, 32.41; S, 43.12; N, 4.45; H, 1.68, in which the NH$_4^+$ are counter ions to balance the charges in the system.

**Conductivity measurements.** The solid sample was finely ground and pressed between two Mylar tapes (sample thickness is about 300 μm and the diameter is around 1.2 cm) at ~1 GPa. The Cu wires were contacted on the pellet surface by carbon paint in a glove box. Thus, a four-point contact was placed at the circumference to define a square. The I–V curves of the pellets were measured in van der Pauw geometry under vacuum at varied temperatures (from 20 K to 320 K) using a commercial Lakeshore Hall System.

**Magnetic studies.** Magnetometry was performed using a SQUID-VSM (Quantum Design). Temperature dependence of the magnetization of the PTC-Fe 2D MOF powder sample was measured in zero-field cooling and field cooling sequence with applied magnetic field of 100 Oe. The magnetic field dependence of magnetization was measured at different temperatures, i.e., 5, 10, 20, 25, and 50 K. Remanent magnetization was measured as a function of temperature in a zero magnetic field, after the sample was cooled down from 350 K to 5 K in a magnetic field of 1000 Oe.

**Data availability.** The data that support the findings of this study are available from the corresponding author on reasonable request.

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

## Acknowledgements

This work was financially supported by the ERC Grant on 2DMATER, EU Graphene Flagship, SPP 1928 (COORNET) and the German Science Council. We acknowledge the cfaed (Center for Advancing Electronics Dresden). We also thank Beamline BL14W1 at the Shanghai Synchrotron Radiation Facility (SSRF) for providing the beamtimes to carry out the XAS measurements. We acknowledge Dresden Center for Nanoanalysis (DCN) at TUD and Dr. Petr Formanek (Leibniz Institute for Polymer Research, IPF, Dresden) for the use of facilities, and we like to appreciate Prof. Stuart Parkin, Dr. Binghai Yan, Dr. Reinhard Berger and Mr. Chi Xu for the helpful discussion.

## Author contributions

X.L.F. and R.H.D. conceived and designed the experiments. R.H.D. synthesized the 2D MOFs, and conducted structure, composition and property characterizations. R.H.D. and M.C.W. synthesized the PTC precursors. Z.T.Z., R.H.D. and S.Q.Z. contributed to the magnetic measurements and analysis. Z.T.Z., S.Q.Z., R.H.D., Z.Z. and S.M. contributed to the conductivity measurements and analysis. D.T., Y.S., W.S. and C.F. performed the theoretical simulation of the 2D MOFs. P.A. and C.F. contributed to the Mössbauer spectroscopy measurements. Z.Q.L., R.H.D. and E.Z. carried out the HRTEM measurements. F.L. performed the X-adsorption spectroscopy experiments. R.H.D. and X.L. F. co-wrote the paper. All authors discussed the results and commented on the manuscript.

## Additional information

**Competing interests:** The authors declare no competing interests.

