## [Peer Review File · Nature Communications]

Reviewers' comments:

Reviewer #1 (Remarks to the Author):

The work has shown that 2D MOF can show ferromagnetism with relatively high Curie temperature. It is of interest and importance for organic spintronics. The paper has claimed that Fe is the origin of the ferromagnetism. However, more details, such as whether only Fe contributes to the magnetism or defects can also contribute to the ferromagnetism. How is the exchange coupling formed leading to ferromagnetic ordering at 20 K. Hence, the paper can be accepted after major revision. The details are shown below:

1. What is the magnetic behavior of PTC without Fe, compared with PTC can help understand the magnetism mechanism of PTC-Fe
2. EDX and XPS to obtain the exact composition of PTC-Fe, which can provide the information whether there are vacancies, which may be another origin of ferromagnetism in PTC-Fe. More discussions should be added.
3. For XAS, Fourier transformation diagram should be provided and proper discussion on the FT should be added since from XANES, the difference as shown in the graph is not clear.
4. For the credit of other works on the magnetism of MOF, more references for magnetic MOF should be added, such as L. Shen et al. *J. Am Chem. Soc.*, 134, 17286 (2012),
5. For the R-T curve, the author has provide R-1/T. $R-(1/T)^{1/2}$ and $R-(1/T)^{1/4}$ should also be provided, which help understand the transport mechanism as seen, Y.R. Wang et al. *Chemistry of Materials* 27 (4), 1285-1291 (2015).
6. Fig. S10-S12, the spin-up and spin down DOS are wrong, where is the spin down DOS.
7. Can the author provide spin density distribution for DFT, which clearly shows that major magnetic moment is from Fe and also whether other elements contribute to the magnetism.
8. Why does the material have relatively high Curie temperature and propose the mechanism for the formation of ferromagnetic ordering.

Reviewer #2 (Remarks to the Author):

The manuscript by Feng and coworkers reports on ferromagnetism at low-temperature (< 20 K) and impressive electrical conductivity at room-temperature (~ 10 S/cm) in a coronene-based MOF. While all it looks very interesting and appealing for a border scientific community, a number of issues raise up which need to be clarified before consideration for publication in *Nature Communications*.

1. The aspects of novelty of present study needs a better description over already reported work, duly referred, ref. 22-24. Also, in view of the 'spintronics' flavor, the authors should briefly refer the metal-organic/molecular semiconductors exhibiting long-range magnetic order even at high-temperatures (*Nature Communication* 5, 3079 (2014) and *J. Phys. Chem. Lett.* 7, 4988 (2016)) likewise they have mentioned Heuslar compounds and DMSc.

Since clear cut saturation magnetization is not observed in the hysteresis plots below 20 K (Figure 5a), the claim of 'ferromagnetic behavior' is an issue. There is a large fraction of paramagnetic moments, as if, ferromagnetic domains are present in a paramagnetic matrix. Figure 5b should be replaced by Figure S15b and vice versa. Apparently, ferromagnetic behavior is not collinear and strongly anisotropic (25% shift in the AB stack) and the authors should comment.

The magnetic ground state in Figure 5c is for single-layer and without inclusion of all Fe atoms. For a better understanding of the long-range magnetic order in the material (within-layer and across-layers), the authors are strongly suggested to present at least 2-3 layers view which can be extracted from DFT data (Figure S13), for example magnetization density plot.

Why the spin-state of Fe(III) is $S=3/2$ not $S=1/2$? A more theoretical insight will be useful.

2. Similar to ref. 25, another important design concept to improve the electrical conductivity of MOF should be mentioned (*J. Phys. Chem. Lett.* 7, 2945 (2016)). The electrical conductivity value is noticeable in the domain of MOFs and I am not sure about the 'record'! The authors can avoid repeating the experimental part in results section.

What is the standard deviation in such electrical measurements and how reliably the value can be reproduced across different batches? Better to mention electrical conductivity of ~ 10 S/cm. Since the material is a 2D framework, electrical conductivity is expected to be anisotropic influencing the overall magnetic order. Did the authors try measuring electrical conductivity in-plane and across-plane geometry in a thicker pellet, for example by simple two-probe?

Also, instead of dispersed-state, solid-state UV-vis data could provide better estimate of optical band-gap which can be correlated with the electronic structure calculations.

3. Inclusion of an XPS survey scan in main text is barely meaningful. The authors are suggested to present Figure S6b and S6c in the main text with spectral fitting of Fe2p.

From the Figure S6c, there seems to be majorly two types of S having different chemical environments, one can be a signature of bound S (Fe-S bond) showing S2p (3/2) signal at ca. 161.5 eV. What about the other major S2p (3/2) signal at ca. 163 eV? Is there any possibility of interlayer S-S linkage (the PXRD peaks are really broad!) leading to electronic mobility paths across the material along with van der Waals stacking? There is a significant overlap of DOS between S and Fe in the bulk band structure (Figure S13).

4. I am surprised that the authors have overlooked an earlier DFT prediction on the electronic structure calculations of coronene-based MOFs (PCCP 18, 25227 (2016)). To be more specific, it the hybridization of S pz-orbital (also C pz-orbital) with dyz and dxz orbitals of Fe.

5. Instead of referring the synthesis of such specific ligand (the authors have wrongly cited 42 instead of 43) the authors are encouraged to briefly describe the method for a broad audience.

Reviewer #3 (Remarks to the Author):

The manuscript describe a new 2D-MOF using a recently reported perthiolated coronene as ligand and planar iron-bis(dithiolene) as linkage to realize π -d conjugation. The MOF powder exhibits a high electrical conductivity of 10.3 S cm⁻¹ at 300 K and ferromagnetic ordering below 20 K. These values are good, but only in the same range as existing other materials. In addition, there are some shortcomings that make the manuscript not appropriate for publication in Nature Communications as it stands now.

1. It is known to all the magnetic properties and the conductivity are really sensitive to impurities. However, there is no elemental analysis of the title compound, meaning that there is essentially no proof of bulk purity. In other words, it is uncertain that the material is of sufficient quality, meaning that the properties can also not be trusted at this point. This is especially true of Fe-containing materials, where ferrite is often an impurity that shows up as a ferromagnetic response. Similar impurities may also be responsible for increasing conductivity, especially at the boundaries of the MOF particles. Also, there is nothing in terms of modelling the magnetic behavior. It is unexpected to see such behavior in this geometry, the authors should attempt how they reconcile this behavior with the potential impurities, with electrical conductivity (unusual for a conductor to display ferromagnetism), and with the structure.

2. As shown in Figure 1b and 1c, the crystallinity of the sample is relatively poor, and the author only indexed three diffraction peaks, so the simulated packing mode is primitive and unacceptable. The author identified the peak at $2\theta=26^\circ$ corresponding to the π - π distance of ~ 3.9 Å. However, the PXRD peak at $2\theta=26^\circ$ is quite broad and seems not in accordance with the simulated data. Also in Figure 1e, it is hard to tell whether it is a vertical side view based on sample with poor crystallinity.

3. The novelty of this work is worth discussing, here the author combined the magnetic property and the electronic conductivity simply and they claimed too much about enormous potential for application in spintronics. As we all know that one important design strategy for spintronics is that using weak spin-orbit interactions materials, such as organic materials due to carbon's lower atomic number and wider band gaps. Therefore, MOFs containing metal atoms are not a good candidate for spintronics.

4. The BET surface area was calculated to be 210 (± 5) m²/g here. It is recommended that the author should point out where the porous property given that the pores are likely occupied by the compensating cations (NH₄⁺).

5. For Figure S8, it is not appropriate to determine the optical band gap by UV-Vis absorption spectra of an insoluble powder in ethanol. This is an all-too-frequent mistake that should not be perpetuated in a journal such as Nature Communications. There is a good primer on these misconceptions called "Mind the gap" by JL Bredas that I highly recommend.

Detailed Responses to the comments of the Reviewers

Reviewer #1:

General Comment: The work has shown that 2D MOF can show ferromagnetism with relatively high Curie temperature. It is of interest and importance for organic spintronics. The paper has claimed that Fe is the origin of the ferromagnetism. However, more details, such as whether only Fe contributes to the magnetism or defects can also contribute to the ferromagnetism. How is the exchange coupling formed leading to ferromagnetic ordering at 20 K. Hence, the paper can be accepted after major revision. The details are shown below:

Response: We appreciate the Reviewer#1 for the encouraging comments and the positive recommendation for publication after revision. According to your valuable suggestions, we have further explored the ferromagnetism mechanism and supplied the additional data about the contrast samples. We hope that our revision addresses your concerns adequately.

Comment 1: What is the magnetic behavior of PTC without Fe, compared with PTC can help understand the magnetism mechanism of PTC-Fe.

Response: We appreciate your constructive suggestions. We researched the ligand PTC and also other starting chemicals (pristine coronene) by SQUID, which indicated that the Fe centers mainly contributed to the magnetic ordering. This experimental result was also supported by additional theoretical calculation about the spin density distribution.

Figure R1 | Temperature dependent remanent magnetization of ligand PTC (red curve), pristine coronene (yellow curve) and PTC-Fe MOF (blue curve).

Figure R1 shows the temperature dependent remanent magnetization of ligand PTC (red curve), pristine coronene (yellow curve) and PTC-Fe MOF (blue curve), which reveals that the complexes by the coordination of PTC and Fe could behave strong ferromagnetism below 20 K. As contrast, ligand PTC and pristine coronene contribute ignorable magnetic ordering even at 5 K. Thus, the magnetic coupling is major relative to the Fe atoms.

Figure R2 | Spin density distribution of AB-stacking PTC-Fe MOF with 25% shifting in X and Y directions between neighboring layers.

Figure R2 shows the spin density distribution of AB-stacking PTC-Fe MOF, which further suggests that the Fe atoms provide major contribution to the magnetic moment compared with the C and S atoms, associated with localized d electrons.

The above additional experimental and theoretical results have been provided in Figures S15 and 16 in the SI, respectively.

Comment 2: EDX and XPS to obtain the exact composition of PTC-Fe, which can provide the information whether there are vacancies, which may be another origin of ferromagnetism in PTC-Fe. More discussions should be added.

Response: According to the suggestions, in the revised manuscript, we provide the detailed discussion about XPS results as well as other compositional analysis. Quantitative analysis of the XPS spectra provides a Fe:S ratio of approximately 0.98:4, which is in consistency with the expected PTC-Fe model (Figure S6). In addition, the elemental analysis of PTC-Fe powder sample was carried out to provide a solid proof of the composition and unambiguously proved the purity: C, 32.41; S, 43.12; N, 4.45; H, 1.68 (Anal. Calcd. for $[\text{Fe}_3\text{C}_{24}\text{S}_{12}]^{3-} \cdot (\text{NH}_4^+)_3$: C, 32.19; S, 42.92; N, 4.69; H, 1.34.), in which the NH_4^+ are counter ions to balance the charges in the system. Moreover, FT-IR analysis of the PTC-Fe MOF revealed that the peak of thiol groups vanished in the PTC-Fe, suggesting that the thiol groups were efficiently coordinated to Fe ions to form iron bis(dithiolene) linkers. Additionally, the combination of XANES and EXAFS (Figures 2a and b, discussion seen below), ^{57}Fe Mössbauer spectra (Figures 2c and d) and high-resolution Fe XPS spectrum (Figure S6b) displayed that only one kind of Fe ions (Fe^{3+}) exist in the MOF sample and confirmed the iron-bis(dithiolene) coordination geometry with the assistance of TEM (Figures 1c-e) and XRD (Figure 1b) investigations. Therefore, the above structural and compositional analysis evidenced the efficient coordination between PTC and Fe ions in this PTC-Fe MOF. As a result, the magnetic coupling should mainly arise from the magnetic moment of the metal coordination complexes but not from the uncoordinated vacancy sites.

Nevertheless, we also agree with the crucial comment of Reviewer#1 on the possible existence of defects. Here, we investigated a polycrystalline framework material, which features structural complexity and possible defects generated from the grain boundaries, the crystallite tilting and the

edges as a result of heterogeneity. On the one hand, perfect crystalline magnetic materials (bulks and films) without defects are always expected and may provide the possibility for unambiguous fundamental understanding of the structure-property relationship, which remain challenging and appealing for the physicists and the MOF community. On the other hand, controlling or engineering the defects is also an intriguing approach to introduce multi-functions for MOF materials (Angew. Chem. Int. Ed. 2015, 54, 7234-7254).

The additional discussion about the defect influence has been added in the revised manuscript (Page 13). The element analysis results have been added in the SI (Page 3)

Comment 3: For XAS, Fourier transformation diagram should be provided and proper discussion on the FT should be added since from XANES, the difference as shown in the graph is not clear.

Response: Thanks for the constructive comments. We have added the FT EXAFS radial distribution curve of PTC-Fe in Figure 2b and Figure S7 in the SI as well as the corresponding discussion in the revised main text (Page 6).

Figure R3 | Fourier transform of the EXAFS at Fe K-edge of synthetic PTC-Fe MOF as well as Fe₂O₃ and TTB-Fe as the contrast samples.

Figure R3 shows the Fourier transform of the κ -edge extended X-ray absorption fine structure (EXAFS) spectra of PTC-Fe as well as Fe₂O₃ and TTB-Fe (synthesized by 1,2,4,5-tetrathiolbenzene and iron(III) ions) as contrast samples. The EXAFS spectra present a predominant peak in PTC-Fe (red curve in Figure R3), which is originated from the nearest-neighboring sulfur coordination shell around the Fe atoms. Based on this peak, Fe-S distance is calculated to be ~ 2.23 Å. Besides this first-shell interaction, the second-shell atomic interaction was also observed, based on which distance of Fe-C of ~ 3.39 Å was calculated. These bond lengths are very close to those of the contrast sample TTB-Fe (blue curve in Figure R3), which are well consistent with the iron-bis(dithiolene) geometry. However, another contrast sample Fe₂O₃ exhibits two predominant peaks at ~ 1.44 Å and ~ 2.57 Å, which arise from Fe-O and Fe-Fe bonds, respectively. Therefore, the XANES (Figure 2b) and EXAFS spectra of PTC-Fe and the contrast experiments provide further proof on the formation of planar iron-bis(dithiolene) complexes via the coordination of PTC and Fe ions. Moreover, there is no metal oxides such as Fe₂O₃ and FeO identified in the PTC-Fe MOF.

Comment 4: For the credit of other works on the magnetism of MOF, more references for magnetic MOF should be added, such as L. Shen et al. *J. Am Chem. Soc.*, 134, 17286 (2012)

Response: We have added this very important literature in our revised manuscript.

In addition, we also cited other important experimental and theoretical reports about the magnetism of metal-organic frameworks.

28. Jeon, I.-R., Negru, B., Van Duyne, R.P. & Harris, T.D. A 2D semiquinone radical-containing microporous magnet with solvent-induced switching from $T_c = 26$ to 80 K. *J. Am. Chem. Soc.* **137**, 15699-15702 (2016).

44. Chakravarty, C., Bikash Mandal, B. & Sarkar, P. Coronene-based metal-organic framework: a theoretical exploration. *Phys. Chem. Chem. Phys.* **18**, 25277-25283 (2016).

53. Friedländer, S. et al. Linear chains of magnetic ions stacked with variable distance: ferromagnetic ordering with a Curie temperature above 20 K. *Angew. Chem. Int. Ed.* **55**, 12683-12687 (2016).

54. Zhou, J. & Sun, Q. Magnetism of phthalocyanine-based organometallic single porous sheet. *J. Am. Chem. Soc.* **133**, 15113-15119 (2011).

55. Lannes, A. et al. Room temperature magnetic switchability assisted by hysteretic valence tautomerism in a layered two-dimensional manganese-radical coordination framework. *J. Am. Chem. Soc.* **138**, 16493-16501 (2016).

56. Espallargas, G.M. & Coronado, E. Magnetic functionalities in MOFs: from the framework to the pore. *Chem. Soc. Rev.* **47**, 533-557 (2018).

61. Shen, L. Origin of long-range ferromagnetic ordering in metal-organic frameworks with antiferromagnetic dimeric-Cu(II) building units. *J. Am. Chem. Soc.* **134**, 17286-17290 (2012).

Comment 5: For the R-T curve, the author has provide R-1/T. R-(1/T)^{1/2} and R-(1/T)^{1/4} should also be provided, which help understand the transport mechanism as seen, Y.R. Wang et al. *Chemistry of Materials* 27 (4), 1285-1291 (2015).

Response: Following the suggestion from the reviewer, we further investigated the charge transport mechanism by Efros-Shklovskii variable range hopping (ES-VRH) model and Mott variable range hopping (Mott-VRH) model expressed by $\sigma(T) = \sigma_0 \exp[-(T_0/T)^s]$ (where σ_0 is a prefactor, T_0 the activation temperature and s an exponent that defines the different hopping transport regimes).

Figure R4 shows the plot of $\ln(\sigma)$ versus $T^{-1/4}$ over the temperature region of 65-130 K, which is well fitted to the Mott-VRH model. This result suggests that the hopping process should be the critical step for charge transport in the PTC-Fe 2D MOF due to the boundaries between the crystallites.

Figure R4 | Plot of $\ln(\sigma)$ versus $T^{-1/4}$ over the temperature region 65-130 K.

We have added the additional data in Figure 4b and cited this important literature in the revised manuscript.

52. Wang, Y. et al. Ferromagnetism and crossover of positive magnetoresistance to negative magnetoresistance in Na-doped ZnO. *Chem. Mater.* **27**, 1285-1291 (2015).

Comment 6: Fig. S10-S12, the spin-up and spin down DOS are wrong, where is the spin down DOS.

Response: We are sorry that we did not provide clear description about Figures S10-S12 in our previous version. Here, we integrated them in one figure and made them concise, as shown in Figure R5.

Figure R5 | Band structure of single layer PTC-Fe. a, Band structure of PTC-Fe with the spin-down state of the Fe atoms. **b,** Band structure of PTC-Fe with the spin-up state of the Fe atoms. **c,** Band structure of PTC-Fe with the spin-down and spin-up states of the Fe atoms. **d,** Total DOS of the system (spin-up and spin down).

The calculation for the density of state (DOS) has been carried out using the VASP

program. The plotting of the spin-up and spin-down DOS has been performed by p4vasp visualization program. The p4vasp program allows to see the local projections of DOS and band structure for a certain atom (e.g. Fe or S) and to focus on the visualization of the targeted orbital separately (e.g. px, py, pz, dxy, dyz etc) as well as the total DOS composed of spin-up and spin-down systems.

Figure R5a presents only the spin-down state of the Fe atoms in single-layer system. It means that, in the input file for VASP, we have specified only spin-down calculations for Fe. In this case, the spin-down state reveals a band gap of ~1 eV. Similarly, Figure R5b only shows the spin-up state of the Fe, which also suggests a band gap of ~1 eV for the Fe spin-up system. While Figure R5c shows the spin-up and spin-down states of the Fe atoms, revealing a band gap of ~0.2 eV. The total DOS composed of the contribution from all atoms (Figure R5c), which displays a rather narrow band gap of ~0.2 eV for the single-layer system. It implies that the spin-up states for this system mainly present very close to the Fermi level. Definitely, the calculation of a band gap for the whole system needs the contribution from both the spin-up and spin-down components.

The re-arranged figure and the additional discussion have been added in the SI (Figure S11).

Comment 7: *Can the author provide spin density distribution for DFT, which clearly shows that major magnetic moment is from Fe and also whether other elements contribute to the magnetism.*

Response: According to the constructive comments from the reviewer, we have carried out the spin density distribution of PTC-Fe by DFT calculation, as shown in Figure R2, which shows the major magnetic moments from Fe atoms.

Comment 8: *Why does the material have relatively high Curie temperature and propose the mechanism for the formation of ferromagnetic ordering.*

Response: We are sorry that we provided a rough discussion about the mechanism of ferromagnetic ordering for the PTC-Fe MOF in our previous manuscript. Here, we attempted to achieve an understanding on the ferromagnetic coupling in the revised manuscript.

As mentioned, with the theoretical calculation of spin density distribution (Figure R2) and the contrast SQUID measurements on the precursors (coronene and PTC in Figure R1), we disclosed that the Fe atoms contribute to the major spins for the magnetic coupling in PTC-Fe. Next, we further explored how the magnetic coupling appeared between the localized spins on Fe atoms. As is defined, this layer-stacked PTC-Fe MOF comprises of conjugated coronene as ligands and planar iron-bis(dithiolene) as linkages, enabling strong conjugation of π electrons, which has a critical impact on the magnetic properties of the lattice. In particular, we observed the unexpected emergence of a relatively high-temperature (<20 K) ferromagnetic semiconducting ground state. Such magnetic ground state exhibits a positive exchange energy (E_{ex}) of 1.22 meV ($E_{\text{ex}} = E_{\text{AFM}} - E_{\text{FM}}$, *J. Am. Chem. Soc.* **133**, 15113-15119 (2011)), which results from the unique hybridization between the d/p orbitals of Fe, the coronene core, and the Fe-bis(dithiolene) nodes. Notably, the magnetic

ground state of the system at 0 K would be FM if $E_{\text{ex}} > 0$, and AFM if $E_{\text{ex}} < 0$. Therefore, the calculated positive E_{ex} of 1.22 meV disclosed the intrinsic ferromagnetic behavior in PTC-Fe. In addition, the simulated Curie temperature of PTC-Fe is ~ 16 K by employing Ising model (calculation method seen in SI, *J. Am. Chem. Soc.* **133**, 15113-15119 (2011); *Chem. Sci.* **8**, 2859-2867 (2017); *J. Phys. Chem. C* **2018**, **122**, 1846-1851), which further supports the SQUID measurements on the magnetic ordering with a blocking temperature of ~ 15 K (Figure 5b in main text).

However, given the distance between the neighboring lateral Fe atoms from ~ 0.54 to ~ 1.2 nm in PTC-Fe MOF, direct exchange between the d orbitals of Fe is relatively weak, and thus cannot be responsible for the strong FM coupling between the intralayer Fe atoms. Additionally, the layer distance is ~ 0.38 nm and thus the distance between the neighboring interlayer Fe atoms is ~ 0.5 nm, also leading to the relatively weak direct exchange interaction. Thus, we proposed that the magnetic coupling was induced by indirect exchange interaction (*J. Am. Chem. Soc.* **137**, 15703-15711 (2015); *J. Am. Chem. Soc.* **137**, 15699-15702 (2016); *J. Am. Chem. Soc.* **139**, 4175-4184 (2017); *Chem. Sci.* **8**, 2859-2867 (2017); *Phys. Chem. Chem. Phys.* **18**, 25277-25283 (2016). *J. Am. Chem. Soc.* **138**, 16493-16501 (2016). *J. Phys. Chem. Lett.* **7**, 4988-4995 (2016)). Namely, the localized spin moments of the Fe atoms polarize the delocalized p orbital through exchange interaction. DOS plots indicate that d_{xz}/d_{yz} orbitals of Fe participate in the hybridization of p orbitals (Figures S10 and S12), leading to strong exchange interaction between the localized d orbitals and the p orbitals, and hence FM coupling. This indirect exchange through delocalized electrons is similar to the Zener d - p exchange (*Science* **287**, 1019-1022 (2000); *Phys. Rev.* **81**, 440-444 (1951)) or the Ruderman-Kittel-Kasuya-Yosida (RKKY) (*Nat. Commun.* **8**, 15388 (2017); *Nat. Mater.* **4**, 173-179 (2005); *Phys. Rev.* **96**, 99-102 (1954); *Prog. Theor. Phys.* **16**, 45-57 (1956); *Phys. Rev.* **106**, 893-898 (1957)) mechanism.

We have added the fundamental understanding on the nature of the ferromagnetic coupling in the Discussion section of our revised manuscript (Pages 12-13) and hope our additional efforts appropriately address your concerns.

Reviewer #2:

General Comment: *The manuscript by Feng and coworkers reports on ferromagnetism at low-temperature (< 20 K) and impressive electrical conductivity at room-temperature (~10 S/cm) in a coronene-based MOF. While all it looks very interesting and appealing for a border scientific community, a number of issues raise up which need to be clarified before consideration for publication in Nature Communications.*

Response: We appreciate the Reviewer#2 for the positive comments that this work “looks very interesting and appealing for a border scientific community”. We are sorry for the inadequate presentation of data regarding structural characterizations and magnetic mechanism discussion in the previous submission.

We hope that our additional efforts in the revised manuscript appropriately address your concerns.

Comment 1: *The aspects of novelty of present study needs a better description over already reported work, duly referred, ref. 22-24.*

Response: As suggested by the reviewer, we have highlighted the important contribution of Ref 22-24 in our revised manuscript (Page 3):

“Immobilizing redox-active ligands with mixed-valences, such as 2,5-dihydroxybenzoquinone, into the backbones of iron(III) complexes could generate long-range charge delocalization and strong magnetic exchange (*J. Am. Chem. Soc.* **137**, 15703-15711 (2015); *J. Am. Chem. Soc.* **137**, 15699-15702 (2016); *J. Am. Chem. Soc.* **139**, 4175-4184 (2017)), leading to high conductivity ($\sim 10^{-4}$ - ~ 0.2 S cm^{-1}) and high-temperature magnetic ordering (Curie temperature even could reach 105 K).”

Comment 2: *Also, in view of the ‘spintronics’ flavor, the authors should briefly refer the metal-organic/molecular semiconductors exhibiting long-range magnetic order even at high-temperatures (Nature Communication 5, 3079 (2014) and J. Phys. Chem. Lett. 7, 4988 (2016)) likewise they have mentioned Heusler compounds and DMSc.*

Response: We have added the description about metal-organic/molecular spintronics and the corresponding literature in the revised Introduction (Page 2):

“Currently, materials employed as ferromagnetic semiconductors generally comprise inorganic solid compounds like Heusler compounds and dilute magnetic semiconductors^{4,6-8} and organic/molecular film semiconductors (*Nature* **427**, 821-824 (2004); *Nat. Mater.* **8**, 707-716 (2009); *Nat. Mater.* **9**, 638-642 (2010); *Nature* **503**, 504-509 (2013); *Nat. Commun.* **5**, 3079 (2014); *J. Phys. Chem. Lett.* **7**, 4988-4995 (2016); *Nat. Phys.* **13**, 994-998 (2017); *Nat. Phys.* **13**, 894-899 (2017))..... In the case of molecular film semiconductors, their electronic and magnetic properties can be tuned to a much higher degree, due to the structural diversities of organic monomers and functional groups that can be employed. Notably, the molecular semiconductors such as vanadium-tetracyanoethylene (*Nat. Mater.* **9**, 638-642 (2010)) and metal-phthalocyanine (*Nature* **503**, 504-509

(2013); *Nat. Commun.* **5**, 3079 (2014); *J. Phys. Chem. Lett.* **7**, 4988-4995 (2016)) have exhibited strong magnetic coupling at high temperature, which are considered as an emerging class of spin transport media with an extraordinarily long spin lifetime due to their carbon-based light-atom compositions. Nevertheless, the low mobility and complex transport properties in molecular semiconductors still hinder their practical applications for spintronics.”

In addition, another literature about the long-range ferrimagnetic order in molecular metal phthalocyanine film on Au substrate via Ruderman-Kittel-Kasuya-Yosida exchange interaction has been cited in our revised manuscript.

“59. Girovsky, J. et al. Long-range ferrimagnetic order in a two-dimensional supramolecular Kondo lattice. *Nat. Commun.* **8**, 15388 (2017).”

Comment 3: *Since clear cut saturation magnetization is not observed in the hysteresis plots below 20 K (Figure 5a), the claim of ‘ferromagnetic behavior’ is an issue. There is a large fraction of paramagnetic moments, as if, ferromagnetic domains are present in a paramagnetic matrix. Figure 5b should be replaced by Figure S15b and vice versa.*

Response: We appreciate the constructive suggestions from the reviewer. In fact, we offered similar explanation in our previous manuscript (Page 11) “These observations point to nanoscale ferromagnetism with an average blocking temperature of about 15 K,” which is due to its polycrystalline feature. In our revised manuscript, we further stressed that the PTC-Fe indeed presents ferromagnetic exchange interactions within nanoscale magnetic clusters (Abstract). Following the comment, we also corrected Figure 5b in the revised version, as shown in Figure R6b.

Figure R6 | Magnetic properties of PTC-Fe. **a**, Magnetizations as functions of applied magnetic field (H) measured at different temperatures. **b**, Magnetization of PTC-Fe as a function of temperature measured in a 100 Oe field under field-cooled (FC) and zero-field-cooled (ZFC) conditions. Inset: Temperature dependent remanent magnetization.

Comment 4: *Apparently, ferromagnetic behavior is not collinear and strongly anisotropic (25% shift in the AB stack) and the authors should comment. The magnetic ground state in Figure 5c is for single-layer and without inclusion of all Fe atoms. For a better understanding of the long-range magnetic order in the material (within-layer and across-layers), the authors are strongly suggested to present*

at least 2-3 layers view which can be extracted from DFT data (Figure S13), for example magnetization density plot.

Response: Following the comments from the reviewer, we carried out the theoretical calculation of spin density distribution of AB-stacking PTC-Fe. Figure R7 shows that the Fe atoms provide major contribution to the magnetic moment in comparison with the C and S atoms, associated with localized d electrons.

Figure R7 | Spin density distribution of AB-stacking PTC-Fe MOF with 25% shifting in X and Y directions between neighboring layers.

However, given the distance between the neighboring lateral Fe atoms from ~ 0.54 to ~ 1.20 nm in PTC-Fe MOF, direct exchange between the d orbitals of intralayer Fe atoms is relatively weak, and thus cannot be responsible for the strong ferromagnetic coupling between the Fe atoms. Additionally, the layer distance is ~ 0.38 nm and the distance between the neighboring interlayer Fe atoms is ~ 0.5 nm, also leading to relatively weak direct exchange interaction. Thus, we propose that the magnetic coupling is induced by indirect exchange interaction (*J. Am. Chem. Soc.* **137**, 15703-15711 (2015); *J. Am. Chem. Soc.* **137**, 15699-15702 (2016); *J. Am. Chem. Soc.* **139**, 4175-4184 (2017); *Chem. Sci.* **8**, 2859-2867 (2017); *Phys. Chem. Chem. Phys.* **18**, 25277-25283 (2016). *J. Am. Chem. Soc.* **138**, 16493-16501 (2016). *J. Phys. Chem. Lett.* **7**, 4988-4995 (2016)). Namely, the localized spin moments of the Fe atoms polarize the delocalized p orbital through exchange interaction. That is because this layer-stacked PTC-Fe MOF comprises of fully conjugated planar structure, enabling strong hybridization between the d/p orbitals of Fe, the coronene core, and the Fe-bis(dithiolene) nodes, which has a critical impact on the magnetic properties of the lattice. DOS plots also indicate that d_{xz}/d_{yz} orbitals of Fe participate in the hybridization of p orbitals (Figures S10 and S12), leading to strong exchange interaction between the localized d orbitals and the p orbitals, and hence ferromagnetic coupling. This indirect exchange through delocalized electrons is consistent with the Zener d - p exchange (*Science* **287**, 1019-1022 (2000); *Phys. Rev.* **81**, 440-444 (1951)) or the Ruderman-Kittel-Kasuya-Yosida (RKKY) (*Nat. Commun.* **8**, 15388 (2017); *Nat. Mater.* **4**, 173-179 (2005); *Phys. Rev.* **96**, 99-102 (1954); *Prog. Theor. Phys.* **16**, 45-57 (1956); *Phys. Rev.* **106**, 893-898 (1957)) mechanism.

We have added the fundamental understanding on the nature of the ferromagnetic coupling in the Discussion section of our revised manuscript (Pages 12-13) and hope our additional efforts appropriately address your concerns.

Comment 5: *Why the spin-state of Fe(III) is $S=3/2$ not $S=1/2$? A more theoretical insight will be useful.*

Response: The $S = 3/2$ (intermediate spin) ground state of Fe^{3+} is a consequence of the square planar coordination geometry, which leads to a pronounced energetic stabilization of the $3d_{z^2}$ orbital with respect to the $d_{x^2-y^2}$ orbital, and the energy of the d_{z^2} orbital is comparable to the energies of the d_{xy} , d_{xz} and d_{yz} orbitals (*Inorg. Chem.* **47**, 10911-10920 (2008); *Angew. Chem. Int. Ed.* **47**, 1228-1231 (2008); *J. Am. Chem. Soc.* **127**, 5641-5654 (2005)).

The corresponding explanation has been added in the Mössbauer discussion (Page 8).

Comment 6: *Similar to ref. 25, another important design concept to improve the electrical conductivity of MOF should be mentioned (*J. Phys. Chem. Lett.* **7**, 2945 (2016)).*

Response: We have added this important literature in the revised manuscript.

Comment 7: *The electrical conductivity value is noticeable in the domain of MOFs and I am not sure about the 'record'! The authors can avoid repeating the experimental part in results section.*

Response: We feel sorry about this inadequate description about the conductivity results. Here, the compressed pellets of bulk PTC-Fe just exhibited high electrical conductivity by van der Pauw method. We avoided repeating the experimental part in the result section.

Comment 8: *What is the standard deviation in such electrical measurements and how reliably the value can be reproduced across different batches? Better to mention electrical conductivity of ~ 10 S/cm.*

Response: Here, we measured four batches of samples by van der Pauw method and obtained an average value with a standard deviation of 1.2. Following the suggestion from the reviewer, we offered ~ 10 S/cm in our revised version.

Comment 9: *Since the material is a 2D framework, electrical conductivity is expected to be anisotropic influencing the overall magnetic order. Did the authors try measuring electrical conductivity in-plane and across-plane geometry in a thicker pellet, for example by simple two-probe?*

Response: We agree with the reviewer that the conducting 2D framework materials could behave anisotropic transport property. However, our current work presents a kind of polycrystalline bulk materials. The disordered arrangement of the crystallites in the bulk phase and the domain boundaries hindered us to well disclose the anisotropic transport behavior. Currently, we are dedicated to further synthesize single-crystalline 2D MOF film with controlled thickness, which would be a promising candidate to address their intrinsic physical properties.

Following the comments from the reviewer, we simply measured the vertical conductivity of the compressed pellets by two-probe method and obtained a conductivity value of ~ 14 S/cm at room temperature, which is close to the lateral one.

Comment 10: Also, instead of dispersed-state, solid-state UV-vis data could provide better estimate of optical band-gap which can be correlated with the electronic structure calculations.

Response: Following the suggestion from the reviewer, we carried out solid-state UV-Vis spectrum (Figure R3a). Importantly, the electronic absorption features of PTC-Fe MOF extend well into the near-infrared (NIR) range. Such low-energy electronic excitations are common in highly conjugated organic/metal-organic and conducting polymers (*J. Phys. Chem. Lett.* **7**, 2945-2950 (2016); *J. Am. Chem. Soc.* **136**, 8859-8862 (2014); *Nat. Energy* **3**, 30-36 (2018); *J. Phys. Chem. B* **119**, 4788-4794 (2015); *Angew. Chem. Int. Ed.* **28**, 1692-1694 (1989); *Angew. Chem. Int. Ed.* **55**, 708-713 (2016)).

Figure R8 | Solid-state UV-Vis absorption spectrum of PTC-Fe MOF.

The additional UV-Vis spectra have been added in Figure S9 in the SI.

Comment 11: Inclusion of an XPS survey scan in main text is barely meaningful. The authors are suggested to present Figure S6b and S6c in the main text with spectral fitting of Fe2p.

Response: Thanks for the constructive suggestion. We moved XPS curve into SI (Figure S6). Here, we added the Fourier transform of the k -weighted extended X-ray absorption fine structure (EXAFS) spectra and the ^{57}Fe Mössbauer spectrum at 5 K in Figure 2 in the revised main text, which are more important to define the Fe valence, the planar square coordination geometry and the purity in PTC-Fe.

Figure R9 | Compositional characterizations of PTC-Fe. **a**, Normalized XANES spectra at the Fe *K*-edge of PTC-Fe and its reference compounds, including Fe foil, FeO, Fe₂O₃, and TTB-Fe(III). For the PTC-Fe, the energy of adsorption edge (E₀) suggests that the valence state of the Fe ion in the PTC is 3+. **b**, Fourier transform of the EXAFS at Fe *K*-edge of PTC-Fe as well as Fe₂O₃ and TTB-Fe as the contrast samples. **c** and **d**, ⁵⁷Fe Mössbauer spectra at 294 K and 5 K, respectively. Dots and black line correspond to experimental data and calculated spectrum, respectively.

Figure R9a and Figure R9c were already discussed in the previous manuscript.

Figure R9b shows the EXAFS spectra of PTC-Fe as well as Fe₂O₃ and TTB-Fe (synthesized by 1,2,4,5-tetrathiolbenzene and iron(III) ions) as contrast samples. The EXAFS spectra present a predominant peak in PTC-Fe, which is originated from the nearest-neighboring sulfur coordination shell around the Fe atoms. Based on this peak, Fe-S distance was calculated to be ~2.15 Å. Besides this first-shell interaction, the second-shell atomic interaction was also observed, based on which distance of Fe-C of ~3.37 Å was calculated. These bond lengths are very close to those of the contrast sample TTB-Fe, which are well consistent with the iron-bis(dithiolene) geometry. However, another contrast sample Fe₂O₃ exhibits two predominant peaks at ~1.44 Å and ~2.57 Å, which arise from Fe-O and Fe-Fe bonds, respectively. Therefore, the XANES (Figure R9a) and EXAFS spectra (Figure R9b) of PTC-Fe and the contrast experiments provide further proof on the formation of planar iron-bis(dithiolene) complexes via the coordination of PTC and Fe ions. Moreover, no metal oxides such as Fe₂O₃ and FeO were detected in the PTC-Fe MOF.

Figure R9d presents the ⁵⁷Fe Mössbauer spectrum at 5 K, which exhibits magnetic hyperfine splitting suggesting a magnetic ordering transition. We can also exclude iron oxide components from the low-temperature Mössbauer spectrum, due to that the harder iron oxides should be more strongly magnetically ordered at 5 K than that of softer metal-organic components. Therefore, Mössbauer spectra at low temperature also prove no iron oxides purities in the PTC-Fe.

Comment 12: From the Figure S6c, there seems to be majorly two types of S having different chemical environments, one can be a signature of bound S (Fe-S bond) showing S2p (3/2) signal at ca. 161.5 eV. What about the other major S2p (3/2) signal at ca. 163 eV? Is there any possibility of interlayer S-S linkage (the PXRD peaks are really broad!) leading to electronic mobility paths across the material along with van der Waals stacking? There is a significant overlap of DOS between S and Fe in the bulk band structure (Figure S13).

Response: We agree with the reviewer that there is possibility for the formation of intralayer -S-S- bonds in the PTC-Fe via oxidation due to the air-sensitive precursor PTC. This is also one possible reason for the final formation of polycrystalline PTC-Fe MOF. In the high-resolution XPS spectrum in the S 2p region (Figure R10), the high-intensity dual peaks at 161.5 and 162.7 eV derive from the -Fe-S- units while the major dual peaks at 163 and 164.2 eV indeed correspond to the -C-S- units. The weak peaks at 164.5 and 165.7 eV are assigned to a negligible fraction of -S-S- bonds, which is hesitated to prove the formation of interlayer -S-S- linkages in PTC-Fe. We already confirmed that the distance between stacking layer was ~ 0.39 nm by XRD while the S-S bond length is only ~ 0.2 nm.

Figure R10 | High-resolution XPS spectrum in the S 2p region.

Comment 13: I am surprised that the authors have overlooked an earlier DFT prediction on the electronic structure calculations of coronen-based MOFs (PCCP 18, 25227 (2016)). To be more specific, it the hybridization of S pz-orbital (also C pz-orbital) with dyz and dxz orbitals of Fe.

Response: We feel sorry to miss such important theory literature. We have added it in our revised version.

Comment 14: Instead of referring the synthesis of such specific ligand (the authors have wrongly cited 42 instead of 43) the authors are encouraged to briefly describe the method for a broad audience.

Response: We corrected this mistake in the revised version and added a brief synthesis description about the precursor PTC (Page 14).

Reviewer #3:

General Comment: *The manuscript describe a new 2D-MOF using a recently reported perthiolated coronene as ligand and planar iron-bis(dithiolene) as linkage to realize π -d conjugation. The MOF powder exhibits a high electrical conductivity of 10.3 S cm⁻¹ at 300 K and ferromagnetic ordering below 20 K. These values are good, but only in the same range as existing other materials. In addition, there are some shortcomings that make the manuscript not appropriate for publication in Nature Communications as it stands now.*

Response: We appreciate the Reviewer#3 for the critical scientific comments. We are sorry for the inadequate presentation of data regarding structural (such as XRD) and property analysis (such as UV-Vis spectrum) in the previous submission, which have been corrected now in this revised version. Particularly, we excluded the possible oxide impurities influence on the magnetic behavior via the combination of XPS, ⁵⁷Fe Mössbauer spectra and XANES as well as the added Fourier transform of the κ -weighted extended X-ray absorption fine structure (EXAFS) spectra and elemental analysis. We also provided the explanation about the potential of such MOF materials for spintronics and the ferromagnetism mechanism with the support of additional SQUID experiments of contrast samples and theoretical calculation. We hope that our additional efforts appropriately address your concerns.

Yet, despite being a critical aspect for the development of MOF based electronics, the simultaneous realization of high electrical conductivity (> 1.0 S cm⁻¹) and magnetic ordering in MOF are relatively unexplored thus far (*J. Am. Chem. Soc.* 139, 4175 (2017); *J. Am. Chem. Soc.* 137, 15699-15702 (2016); *Angew. Chem. Int. Ed.* 55, 12683-12687 (2016); *J. Am. Chem. Soc.* 137, 15703 (2015)). Apparently, the fundamental understanding on the relationship between its structure and ferromagnetic conducting property remains ambiguous, which is attracting chemists, materials scientists and physicists to contribute enormous efforts.

Comment 1: *It is known to all the magnetic properties and the conductivity are really sensitive to impurities. However, there is no elemental analysis of the title compound, meaning that there is essentially no proof of bulk purity. In other words, it is uncertain that the material is of sufficient quality, meaning that the properties can also not be trusted at this point. This is especially true of Fe-containing materials, where ferrite is often an impurity that shows up as a ferromagnetic response. Similar impurities may also be responsible for increasing conductivity, especially at the boundaries of the MOF particles. Also, there is nothing in terms of modelling the magnetic behavior. It is unexpected to see such behavior in this geometry, the authors should attempt how they reconcile this behavior with the potential impurities, with electrical conductivity (unusual for a conductor to display ferromagnetism), and with the structure.*

Response: Thanks for the critical comments on the purity issue and the ferromagnetism mechanism. To address these concerns from the Reviewer#3, we carried out additional compositional analysis of PTC-Fe, SQUID experiments on contrast samples and theoretical calculation on the spin distribution and Curie temperature, and provided a discussion on the magnetism mechanism.

1. We excluded the possible influence of oxides impurities on the magnetic behavior in this PTC-Fe MOF.

We presented a detailed discussion about XPS and XAS results as well as other compositional analysis in our revised manuscript. Quantitative analysis of the XPS spectra provided a Fe:S ratio of approximately 0.98:4, which is in consistency with the expected PTC-Fe model (Figure S6 in SI). Moreover, FT-IR analysis of the PTC-Fe MOF revealed that the peak of thiol groups vanished in the PTC-Fe, suggesting that the thiol groups were efficiently coordinated to Fe ions to form iron bis(dithiolene) linkers.

Figure R11 | Compositional characterizations of PTC-Fe. **a**, Normalized XANES spectra at the Fe K-edge of PTC-Fe and its reference compounds, including Fe foil, FeO, Fe₂O₃, and TTB-Fe(III). For the PTC-Fe, the energy of adsorption edge (E₀) suggests that the valence state of the Fe ion in the PTC is 3+. **b**, Fourier transform of the EXAFS at Fe K-edge of PTC-Fe as well as Fe₂O₃ and TTB-Fe as the contrast samples. **c** and **d**, ⁵⁷Fe Mössbauer spectra at 294 K and 5 K, respectively. Dots and black line correspond to experimental data and calculated spectrum, respectively.

Synchrotron powder X-ray absorption spectroscopy (XAS) was employed to thoroughly characterize the coordination geometry and iron valence in PTC-Fe (Figure R11a). As a reference, the known one-dimensional coordination polymer (TTB-Fe) comprising 1,2,4,5-tetrathiolbenzene and iron (III) as well as Fe foil, FeO and Fe₂O₃ inorganic solids were also investigated by XAS. Thus, the Fe K-edge X-ray absorption near edge structure (XANES) spectra of PTC-Fe reveal the same coordination geometry as the TTB-Fe. The Fourier transform of the κ -weighted extended X-ray absorption fine structure (EXAFS) spectra of PTC-Fe as well as Fe₂O₃ and TTB-Fe as contrast samples was additionally provided, as shown in Figure R11b. The EXAFS spectra present a predominant peak in PTC-Fe, which is originated from the nearest-neighboring sulfur coordination shell around the Fe atoms. Based on this peak, Fe-S distance was calculated to be ~ 2.23 Å. Besides this first-shell interaction, the second-

shell atomic interactions were also observed, based on which distance of Fe-C of ~ 3.39 Å was calculated (Figure S7 in SI). These bond lengths are very close to those of the contrast sample TTB-Fe, which are well consistent with the iron-bis(dithiolene) geometry. However, another contrast sample Fe_2O_3 exhibits two predominant peaks at ~ 1.44 Å and ~ 2.57 Å, which arise from Fe-O and Fe-Fe bonds, respectively. Therefore, the XANES and EXAFS spectra of PTC-Fe and the contrast experiments provide strong proof on the formation of square planar iron-bis(dithiolene) complexes via the coordination of PTC and Fe ions. Moreover, no metal oxides such as FeO and Fe_2O_3 were detected in the PTC-Fe.

^{57}Fe Mössbauer spectra (Figures R11c and d), which are also very sensitive to the local environment of Fe atom and its valence, also display that only one kind of Fe ions (Fe^{3+}) exist in the MOF sample, coming from the iron-bis(dithiolene) coordination geometry. Thus, we can also exclude iron oxide components from the low-temperature Mössbauer spectrum, due to that the harder iron oxides should be more strongly magnetically ordered at 5 K than that of softer metal-organic components. Therefore, Mössbauer spectra at low temperature also prove no iron oxides purities in the PTC-Fe.

Following the suggestion from the reviewer, the elemental analysis of PTC-Fe was measured to provide a solid proof of the composition and unambiguously proved the purity: C, 32.41; S, 43.12; N, 4.45; H, 1.68 (Anal. Calcd. for $[\text{Fe}_3\text{C}_{24}\text{S}_{12}]^{3-} \cdot (\text{NH}_4^+)_3$: C, 32.19; S, 42.92; N, 4.69; H, 1.34.), in which the NH_4^+ are counter ions to balance the charges in the system.

As a result, the combination of XPS, ^{57}Fe Mössbauer spectra, XANES, EXAFS and elemental analysis confirmed the high purity of PTC-Fe without fraction of iron oxides.

Figure R11 has been adopted as Figure 2 in the revised main text and the elemental analysis has been added in the SI (Page 3).

2. We provided a mechanism description about the ferromagnetic coupling in PTC-Fe with the support from additional theoretical calculation and SQUID experiments of contrast samples.

The magnetic properties of organic ligand PTC and pristine coronene were investigated by SQUID. Figure R12 shows the temperature dependent remanent magnetization of ligand PTC (red curve), coronene (yellow curve) and PTC-Fe MOF (blue curve), which reveals that the complexes by the coordination of PTC and Fe could behave strong ferromagnetism below 20 K. As contrast, ligand PTC and coronene only contribute ignorable magnetic ordering even at 5 K. Thus, this controlled SQUID experiment indicated that the magnetic property in PTC-Fe was major relative to the Fe atoms.

Figure R12 | Temperature dependent remanent magnetization of ligand PTC (red curve), starting chemical coronene (yellow curve) and PTC-Fe MOF (blue curve). Inset: the enlarged image.

To further explore the origin of magnetic moments, the spin density distribution in AB-stacking PTC-Fe MOF was simulated by DFT method. Figure R13 reveals that the spins mainly locate on the Fe centers compared with the C and S atoms, which are associated with localized d electrons. Therefore, the above experimental and theoretical results evidenced that the Fe metal centers contributed the major magnetic moment for the magnetic coupling.

Figure R13 | Spin density distribution of AB-stacking PTC-Fe MOF with 25% shifting in X and Y directions between neighboring layers.

Here, we attempted to achieve a fundamental understanding on how the strong magnetic coupling appears between the spins localized on Fe atoms. As is defined, this layer-stacked PTC-Fe MOF comprises of conjugated coronene as ligands and planar iron-bis(dithiolene) as linkages, enabling strong conjugation of π electrons, which has a critical impact on the magnetic properties of the lattice. In particular, we observed the unexpected emergence of a relatively high-temperature (<20 K) ferromagnetic semiconducting ground state. Such magnetic ground state exhibits a positive exchange energy (E_{ex}) of 1.22 meV ($E_{ex} = E_{AFM} - E_{FM}$, *J. Am. Chem. Soc.* **133**, 15113-15119 (2011)), which results from the unique hybridization between the d/p orbitals of Fe, the coronene core, and the Fe-bis(dithiolene) nodes. Notably, the magnetic ground state of the system at 0 K would be FM if $E_{ex} > 0$, and AFM if $E_{ex} < 0$. Therefore, the calculated positive E_{ex} of 1.22 meV disclosed the intrinsic ferromagnetic behavior in PTC-Fe (Figure 5c in main text). In addition, the simulated Curie

temperature of PTC-Fe is ~16 K by employing Ising model (calculation method seen in SI, *J. Am. Chem. Soc.* **133**, 15113-15119 (2011); *Chem. Sci.* **8**, 2859-2867 (2017); *J. Phys. Chem. C* 2018, **122**, 1846-1851), which further supports the SQUID measurements on the magnetic ordering with a blocking temperature of ~15 K (Figure 5b in main text).

However, given the distance between the neighboring lateral Fe atoms from ~0.54 to ~1.2 nm in PTC-Fe MOF, direct exchange between the *d* orbitals of Fe is relatively weak, and thus cannot be responsible for the strong FM coupling between the intralayer Fe atoms. Additionally, the layer distance is ~0.38 nm and the distance between the neighboring interlayer Fe atoms is ~0.5 nm, also leading to relatively weak direct exchange interaction. Thus, we proposed that the magnetic coupling was induced by indirect exchange interaction (*J. Am. Chem. Soc.* **137**, 15703-15711 (2015); *J. Am. Chem. Soc.* **137**, 15699-15702 (2016); *J. Am. Chem. Soc.* **139**, 4175-4184 (2017); *Chem. Sci.* **8**, 2859-2867 (2017); *Phys. Chem. Chem. Phys.* **18**, 25277-25283 (2016). *J. Am. Chem. Soc.* **138**, 16493-16501 (2016). *J. Phys. Chem. Lett.* **7**, 4988-4995 (2016)). Namely, the localized spin moments of the Fe atoms polarize the delocalized *p* orbital through exchange interaction. DOS plots indicate that d_{xz}/d_{yz} orbitals of Fe participate in the hybridization of *p* orbitals (Figures S10 and S12 in SI), leading to strong exchange interaction between the localized *d* orbitals and the *p* orbitals, and hence FM coupling. This indirect exchange through delocalized electrons is similar to the Zener *d-p* exchange (*Science* **287**, 1019-1022 (2000); *Phys. Rev.* **81**, 440-444 (1951)) or the Ruderman-Kittel-Kasuya-Yosida (RKKY) (*Nat. Commun.* **8**, 15388 (2017); *Nat. Mater.* **4**, 173-179 (2005); *Phys. Rev.* **96**, 99-102 (1954); *Prog. Theor. Phys.* **16**, 45-57 (1956); *Phys. Rev.* **106**, 893-898 (1957)) mechanism.

We have added the description on the nature of the ferromagnetic coupling in the Discussion section of our revised manuscript and hope our additional efforts appropriately address your concerns.

Comment 2: As shown in Figure 1b and 1c, the crystallinity of the sample is relatively poor, and the author only indexed three diffraction peaks, so the simulated packing mode is primitive and unacceptable. The author identified the peak at $2\theta=26\theta$ corresponding to the π - π distance of ~3.9 Å. However, the PXRD peak at $2\theta=26\theta$ is quite broad and seems not in accordance with the simulated data. Also in Figure 1e, it is hard to tell whether it is a vertical side view based on sample with poor crystallinity.

Response: We fully understand the concerns from the reviewer on the crystallinity and the structural determination (such as repeated units and stacking pattern). As one of the common targets, it remains challenging for the community to synthesize highly crystalline or even single-crystalline layer-stacked 2D framework materials and achieve a strongly reliable structure-property relationship.

Here, we synthesized a polycrystalline layer-stacked 2D MOF, which comprises of poly-dispersed nanoscale crystallites. Following the comments from the reviewer, we re-simulated the XRD plots by tuning the position of the counterions, which has a large influence on the layer distance, and then fully assigned the major peaks in the experimental PXRD spectra.

Figure R14 | Experimental and calculated PXRD patterns. Inset: enlarged experimental PXRD peak corresponding to (001) reflection.

Figure R14 shows that the PTC-Fe presents four prominent peaks at $2\theta = 8.8^\circ$, 17.6° , 21.7° and 26.0° , which corresponding to (100), (200), (210) and (001) reflections, respectively. After matching with calculated stacking models, the experimental PXRD evidenced a AB stacking structure with 25% shifting in X and Y directions between neighboring layers for PTC-Fe 2D MOF and the layer distance was calculated as $\sim 3.9 \text{ \AA}$.

The broad peak appearing from $\sim 24^\circ$ to $\sim 27^\circ$ generates from the π - π stacking layers (inset in Figure R14), which is very general for the layer-stacked 2D framework materials but does not imply poor crystallinity, such as 2D MOFs (*J. Am. Chem. Soc.* 136, 8859-8862 (2014); *Nat. Commun.* 6, 7408 (2015); *Angew. Chem. Int. Ed.* 54, 4349-4352 (2015); *J. Am. Chem. Soc.* 139, 19-22 (2017); *Angew. Chem. Int. Ed.* 56, 16510-16514 (2017); *J. Am. Chem. Soc.* 139, 13608-13611 (2017)) and 2D COFs (*Science* 310, 1166-1170 (2005); *Nat. Commun.* 4, 2736 (2013); *Nat. Mater.* 15, 722-726 (2016); *J. Am. Chem. Soc.* 139, 9558-9565 (2017); *J. Am. Chem. Soc.* 139, 6042-6045 (2017)). For instance, $\text{Ni}_3(\text{HITP})_2$ 2D MOF synthesized by Dincă group showed stacking peak from $\sim 26^\circ$ to $\sim 28^\circ$ (*J. Am. Chem. Soc.* 136, 8859-8862 (2014)), $\text{Cu}_3(\text{HITP})_2$ 2D MOF displayed broad (001) peak from $\sim 23^\circ$ to $\sim 31^\circ$ (*Angew. Chem. Int. Ed.* 54, 4349-4352 (2015)), and Cu-BHT 2D MOF reported by Zhu group also exhibited broad (001) peak from $\sim 25^\circ$ to $\sim 30^\circ$ (*Nat. Commun.* 6, 7408 (2015)). Regarding to the 2D COFs, COF-5 reported by Yaghi group showed broad (001) peak from $\sim 24^\circ$ to $\sim 28^\circ$ (*Science* 310, 1166-1170 (2005)), TPB-DMTP-COF synthesized by Jiang group also presented broad stacking peak from $\sim 24^\circ$ to $\sim 28^\circ$ (*Nat. Mater.* 15, 722-726 (2016)), the recently reported salen-based COF by Wang group exhibited broad (001) peak from $\sim 24^\circ$ to $\sim 28^\circ$ (*J. Am. Chem. Soc.* 139, 6042-6045 (2017)), and a viologen-based COF by Trabolsi group presented very broad stacking peak from $\sim 16^\circ$ to $\sim 40^\circ$ (*J. Am. Chem. Soc.* 139, 9558-9565 (2017)).

In addition, it is always challenging for the TEM community to acquire high-resolution structural image of 2D MOFs, due to their high sensitivity to electron beam (*Nat. Mater.* 16, 532-536 (2017); *J. Am. Chem. Soc.* 139, 19-22 (2017); *Angew. Chem. Int. Ed.* 54, 12058-12063 (2015)). In fact, our work indeed offers a high-quality high-resolution TEM image (Figures 1d and e in the main text), which clearly presents nanoscale crystalline domains in PTC-Fe and demonstrates a hexagonal crystalline structure with lattice parameters of $a = b = \sim 1.2 \text{ nm}$, $c = \sim 4 \text{ \AA}$, which are well consistent with PXRD plots.

According to the comments, we have corrected the XRD plots in Figure 1b in the revised main text as well as Figure S4 in SI.

Comment 3: *The novelty of this work is worth discussing, here the author combined the magnetic property and the electronic conductivity simply and they claimed too much about enormous potential for application in spintronics. As we all know that one important design strategy for spintronics is that using weak spin-orbit interactions materials, such as organic materials due to carbon's lower atomic number and wider band gaps. Therefore, MOFs containing metal atoms are not a good candidate for spintronics.*

Response: We agree with the reviewer that the design of organic semiconductors favors the utilization of carbon-based light-atom materials with low spin-orbit coupling (SOC) and long spin-relaxation time (*Nat. Mater.* 8, 707-716 (2009)). We mentioned this in our revised manuscript. However, it does not imply that the spin-orbital coupling should be ignored. Here, we politely disagree with the reviewer that the conducting conjugated 2D MOFs are not a good candidate for spintronics due to their relatively high SOC.

In fact, the role of SOC is much more complex than the reviewer suggests. In this case, as suggested by our collaboration partner Prof. Stuart Parkin (Max Planck Institute of Microstructure Physics, Halle, Germany), operating spintronic devices includes spin injection, transport, spin polarization manipulation, and detection. Indeed, one of the most exciting research topics in spintronics today is what one might call spin-orbitronics where spin-orbit coupling plays a key role, for example, to generate spin-currents via the spin Hall effect, or to generate magnetic anisotropy at interfaces between two magnetic layers or magnetic and non-magnetic layers. Secondly, spin-orbit coupling is not simply linked to atomic number but rather to symmetries and topologies and even very Z materials can exhibit high spin-orbit coupling (e.g. hot spots in Aluminum, see e.g. *Phys. Rev. Lett.* 83, 1211-1214 (1999); *Scientific Reports* 6, 22706 (2016)). Spintronics is about taking advantage of spin-orbit coupling.

In terms of materials development, much of the efforts have been focused on making the ferromagnetic electrode either half metallic or a magnetic semiconductor for efficient spin injection. The evaluation of new materials for the spin transport and manipulation medium has been focused on either low SOC materials for long spin lifetimes (*Nat. Phys.* 13, 994-999 (2017); *Nat. Commun.* 8, 15200 (2017)); or high SOC materials for spin generation and manipulation (*Nat. Phys.* 13, 894-900 (2017); *Nat. Phys.* 9, 284-287 (2013)).

Moreover, many metal-organic molecules with high SOC have been emerging as potential materials for spintronics, such as tris(8-hydroxyquinolinato)aluminium (*Nature* 427, 821-824 (2004)), vanadium-tetracyanoethylene (*Nat. Mater.* 9, 638-642 (2010)), metal-phthalocyanine (*Nature* 503, 504-509 (2013); *Nat. Commun.* 5, 3079 (2014); *J. Phys. Chem. Lett.* 7, 4988-4995 (2016)) and organic-inorganic perovskite $\text{CH}_3\text{NH}_3\text{PbCl}_{3-x}$ (*Nat. Phys.* 13, 894-900 (2017)). Nevertheless, the low mobility and complex transport properties in molecular semiconductors still hinder their practical applications for spintronics.

Currently, some 3D MOFs have shown their potential roles as spintronic materials (*J. Am. Chem. Soc.* **139**, 4175-4184 (2017); *J. Am. Chem. Soc.* **138**, 6583-6590 (2016); *J. Am. Chem. Soc.* **137**, 15703-15711 (2015)) along with a large number of reports on theoretical modeling (*Phys. Rev. B* **85**, 115201 (2012); *Nat. Commun.* **4**, 1471 (2013); *Nano. Lett.* **16**, 2072-2075 (2016); *Phys. Rev. B* **94**, 081102 (2016); *Chem. Sci.* **8**, 2859-2867 (2017)). However, the conductivity in these 3D MOFs is still relatively low. In this respect, with the rising of conducting conjugated 2D MOFs, it is possible to simultaneously achieve high electrical conductivity/charge mobility and long-range magnetic ordering within a single metal-organic material, which makes 2D MOF as a promising candidate for spintronics.

We have mentioned the currently developed metal-organic molecules with high-temperature magnetic coupling for spintronics in our revised introduction (Page 2).

Comment 4: The BET surface area was calculated to be 210 (± 5) m²/g here. It is recommended that the author should point out where the porous property given that the pores are likely occupied by the compensating cations (NH₄⁺).

Response: We appreciate the valuable comment from the reviewer. The counter ions of NH₄⁺ took up the pores in PTC-Fe MOF, leading to the relatively low BET surface area. We have pointed it out in the revised manuscript.

Comment 5: For Figure S8, it is not appropriate to determine the optical band gap by UV-Vis absorption spectra of an insoluble powder in ethanol. This is an all-too-frequent mistake that should not be perpetuated in a journal such as Nature Communications. There is a good primer on these misconceptions called "Mind the gap" by JL Bredas that I highly recommend.

Response: We feel very sorry about the inadequate presentation of the UV-Vis spectra and the rough definition of the optical gap in our previous manuscript. Instead of the dispersed-state, solid-state UV-vis spectrum was carried out. Figure R15 shows the solid-state UV-Vis spectrum of PTC-Fe. Importantly, the electronic absorption features of PTC-Fe MOF extend well into the near-infrared (NIR) range. Such low-energy electronic excitations are common in highly conjugated organic/metal-organic and conducting polymers (*J. Phys. Chem. Lett.* **7**, 2945-2950 (2016); *J. Am. Chem. Soc.* **136**, 8859-8862 (2014); *Nat. Energy* **3**, 30-36 (2018); *J. Phys. Chem. B* **119**, 4788-4794 (2015); *Angew. Chem. Int. Ed.* **28**, 1692-1694 (1989); *Angew. Chem. Int. Ed.* **55**, 708-713 (2016).).

Figure R15 | Solid-state UV-Vis absorption spectrum of PTC-Fe MOF.

The additional UV-Vis spectra have been added in Figure S9 in the SI.

REVIEWERS' COMMENTS:

Reviewer #1 (Remarks to the Author):

I am ok for the current version and agree to publish.

Reviewer #2 (Remarks to the Author):

I have carefully read the rebuttal of the authors and the revised manuscript. Indeed, the rebuttal is nicely done and much Review friendly!

The authors have considered my remarks (as well as other comments from Reviewer 1 and Reviewer 3) during the first review and have taken appropriate action. To my mind the manuscript has improved considerably to a degree where I recommend it for publication in Nature Comm. as is.

The paper will be a useful contribution to the emerging field of magnetic MOFs.

Detailed Responses to the comments of the Reviewers

Reviewer #1:

General Comment: *I am ok for the current version and agree to publish.*

Response: We appreciate the Reviewer#1 for the positive recommendation for publication.

Reviewer #2:

General Comment: *I have carefully read the rebuttal of the authors and the revised manuscript. Indeed, the rebuttal is nicely done and much Review friendly!*

The authors have considered my remarks (as well as other comments from Reviewer 1 and Reviewer 3) during the first review and have taken appropriate action. To my mind the manuscript has improved considerably to a degree where I recommend it for publication in Nature Comm. as is.

The paper will be a useful contribution to the emerging field of magnetic MOFs.

Response: We appreciate the Reviewer#2 for the encouraging comments and the positive recommendation for publication.